# Mitochondrial ABHD11 inhibition drives sterol metabolism to modulate T-cell effector function

Benjamin J. Jenkins[1,19], Yasmin R. Jenkins[1,19], Fernando M. Ponce-Garcia[1], Chloe Moscrop[2], Iain A. Perry[1], Matthew D. Hitchings[1], Alejandro H. Uribe[3], Federico Bernuzzi[3], Simon Eastham[2], James G. Cronin[1], Ardena Berisha[4], Alexandra Howell[5], Joanne Davies[5], Julianna Blagih[6,7], Marta Williams[8], Morgan Marsden[8], Douglas J. Veale[9], Luke C. Davies[1], Micah Niphakis[10], David K. Finlay[11], Linda V. Sinclair[12], Benjamin F. Cravatt[10], Andrew E. Hogan[4], James A. Nathan[13], Ian R. Humphreys[8], Ursula Fearon[14], David Sumpton[3], Johan Vande Voorde[3,15], Goncalo Dias do Vale[16], Jeffrey G. McDonald[16], Gareth W. Jones[2], James A. Pearson[5,20], Emma E. Vincent[17,18,20] & Nicholas Jones[1,20] ✉

α/β-hydrolase domain-containing protein 11 (ABHD11) is a mitochondrial hydrolase that maintains the catalytic function of α-ketoglutarate dehydrogenase (α-KGDH), and its expression in CD4 + T-cells has been linked to remission status in rheumatoid arthritis (RA). However, the importance of ABHD11 in regulating T-cell metabolism and function is yet to be explored. Here, we show that pharmacological inhibition of ABHD11 dampens cytokine production by human and mouse T-cells. Mechanistically, the anti-inflammatory effects of ABHD11 inhibition are attributed to increased 24,25-epoxycholesterol (24,25-EC) biosynthesis and subsequent liver X receptor (LXR) activation, which arise from a compromised TCA cycle. The impaired cytokine profile established by ABHD11 inhibition is extended to two patient cohorts of autoimmunity. Importantly, using murine models of accelerated type 1 diabetes (T1D), we show that targeting ABHD11 suppresses cytokine production in antigen-specific T-cells and delays the onset of diabetes in vivo in female mice. Collectively, our work provides pre-clinical evidence that ABHD11 is an encouraging drug target in T-cell-mediated inflammation.

Activated T-cells undergo extensive metabolic reprogramming to support the biosynthetic and energetic demands upon antigen encounter[1,2]. This anabolic switch supports effector function, providing the necessary precursors for cytokine production and blastogenesis. Failure to appropriately regulate T-cell activation culminates in autoimmunity, whereby augmented T-cell function is driven by alternate or heightened metabolic programmes[3,4]. Increasingly, altered mitochondrial metabolism within the T-cell compartment is being implicated in the onset of autoimmunity. For instance, in rheumatoid arthritis (RA), CD4+ T-cells lack mitochondrial aspartate, which disrupts the regeneration of metabolic cofactors required for ER sensor modification, to become enriched in rough endoplasmic reticulum capable of producing large amounts of TNFα, driving inflammation within surrounding tissues[5]. This arthritogenic phenotype arises from mitochondrial malfunction, whereby mitochondrial DNA damage uncouples oxidative phosphorylation (OXPHOS) from ATP

production[6], whilst defective succinyl-CoA ligase function drives reductive carboxylation[7]. Indeed, TNFα itself is likely to be a major driver of these changes in mitochondrial metabolism[8]. Mitochondrial dysfunction is a feature of many autoimmune conditions, as pathogenic T-cells in systemic lupus erythematosus (SLE) can be characterised by elevated levels of OXPHOS[4,9], whilst their counterparts in multiple sclerosis (MS) display reduced levels of mitochondrial respiration[10]. These distinct modes of mitochondrial dysfunction highlight the metabolic heterogeneity that underpins autoimmunity.

Several leading immunosuppressants target the immunometabolic profile to modulate inflammation[11]. For example, methotrexate has long been a first-line treatment of many autoimmune disorders, inhibiting key aspects of folate metabolism and nucleotide synthesis to limit immune cell function[12]. However, a substantial proportion of autoimmune patients do not respond to methotrexate[13], whilst other treatments are often blighted by debilitating side effects[14]. To this end, novel treatment strategies have been developed. Both tetramerisation of pyruvate kinase[15], and inhibition of ATP synthase[16], have improved disease outcomes in experimental autoimmune encephalomyelitis (EAE) models, harnessing the relationship between metabolism and autoimmunity for therapeutic benefit. Moreover, metabolic modulators such as 2-deoxyglucose and metformin have shown promise alongside traditional treatment strategies in lupus-prone mice[17], which, amongst other studies, has catalysed the repurposing of metformin in SLE[18], RA[19] and multiple sclerosis[20] in clinical trials. Thus, there is substantial precedent for manipulating T-cell metabolism to uncover alternative therapeutic agents that alleviate inflammation.

Molecular profiling has identified α/β-hydrolase domain-containing protein 11 (ABHD11) as one of the genes most associated with remission status in RA[21]. Specifically, reduced ABHD11 expression within patient-derived CD4+ T-cells correlates with improved disease outcome and clinical remission[21]. ABHD11 plays a peripheral role in mitochondrial metabolism, where it regulates the activity of α-ketoglutarate dehydrogenase (α-KGDH), the rate-limiting enzyme within the TCA cycle, by maintaining functional lipoylation of its catalytic subunit, dihydrolipoamide S-succinyltransferase (DLST)[22]. In the absence of ABHD11 function, α-ketoglutarate (α-KG) accumulates and is subsequently converted to 2-hydroxyglutarate (2-HG), which can inhibit the function of α-KG-dependent dioxygenases involved in epigenetic remodelling and hypoxia-inducible factor (HIF) signalling[22]. Whilst 2-HG has previously been shown to suppress cytokine production and cytotoxicity in murine CD8+ T-cells[23], the importance of ABHD11 function in T-cells has yet to be determined.

Here, we show that loss of ABHD11 function limits T-cell cytokine production. Mechanistically, we demonstrate these changes are not attributed to a 2-HG epigenetic axis, rather an augmented oxysterol biosynthesis programme that arises following the accumulation of lactate and acetyl-CoA. To this end, heightened intracellular lactate promotes SREBP2 signalling, which upregulates sterol biosynthetic processes to drive the production of 24,25-epoxycholesterol (24,25-EC) via a mevalonate shunt pathway and subsequent activation of liver X receptor (LXR) signalling. Crucially, ex vivo pharmacological inhibition of ABHD11 suppressed T-cell function in two autoimmune patient cohorts, whilst also attenuating disease activity in a murine model of accelerated type 1 diabetes (T1D) in vivo in female mice. Together, these data demonstrate the potential of modulating ABHD11 function for therapeutic benefit in T-cell-mediated inflammation.

## Results

### ABHD11 is required for murine and human T-cell effector function

Given that reduced expression of ABHD11 within CD4+ T-cells is associated with remission in RA[21], we sought to better understand the role of ABHD11 in T-cell biology. Firstly, we established the degree of ABHD11 protein expression in human CD4+ T-cells, wherein ABHD11 was expressed at low levels in unstimulated T-cells, but became notably upregulated upon TCR-mediated activation (Fig. 1a). To determine the importance of ABHD11 function, human CD4+ T-cells were treated with ML-226, a highly-selective inhibitor that targets the active site serine of ABHD11[24], whilst concomitantly being activated by anti-CD3 and anti-CD28 for 24 h. Here, ML-226 significantly impaired cytokine production, with striking reductions in the release of IL-2, IL-10, IL-17, IFNγ and TNFα (Fig. 1b). We also analysed underlying mRNA levels, where comparable reductions were observed at the gene transcript level (Fig. 1c). Although there were modest reductions in CD25, CD44 and CD69 expression following ABHD11 inhibition, the proportion of cells that upregulate the expression of these markers following TCR stimulation remained similar (Fig. 1d; Supplementary Fig. 1a). Despite the loss of effector function, ABHD11 inhibition did not induce any notable reduction in T-cell size or proliferation (Fig. 1e, f). In line with this, global protein translation, as measured by puromycin incorporation, was not significantly altered by ABHD11 inhibition (Supplementary Fig. 1b). Importantly, cell viability remained intact in T-cells exposed to ML-226 (Supplementary Fig. 1c, d), confirming that the observed loss of effector function is attributable to ABHD11 inhibition rather than compromised cell viability.

We next examined proximal TCR signalling events in the presence or absence of ABHD11 inhibition. Interestingly, nascent phosphorylation events (≤5 min) were unperturbed following ABHD11 inhibition, however, all subsequent time points analysed (≥15 min) evidenced compromised TCR signalling, suggesting that T-cells are unable to sustain proximal TCR signalling events in the presence of ABHD11 inhibition (Fig. 1g). Given that our data suggest ABHD11 inhibition impairs global T-cell function, we next assessed its effect on regulatory T-cells (Treg). Here, polarisation of CD4+ naïve T-cells towards the Treg compartment was significantly reduced by ABHD11 inhibition (Supplementary Fig. 1e). Moreover, Tregs cultured in the presence of ML-226 for 24 h displayed significantly reduced FOXP3 expression (Supplementary Fig. 1f). In order to determine whether pharmacological inhibition of ABHD11 could be replicated using genetic ablation, we developed a series of Jurkat T-cell clones with ABHD11 gene expression knocked down using CRISPR/Cas9 (Fig. 1h). In agreement with pharmacological blockade, ABHD11 knockdown resulted in a significant reduction of IL-2 production upon stimulation with PMA/ionomycin (Fig. 1i).

To further investigate the impact of ABHD11 function on T-cell fitness, we activated murine T-cells in the presence and absence of WWL222, a potent and selective inhibitor targeting ABHD11 in mice[25]. WWL222 is structurally distinct from ML-226, containing a different chemical scaffold, and, importantly, targets an alternate region of ABHD11[24]. Specifically, murine T-cells were polarised in vitro towards distinct T-cell lineages in the presence and absence of WWL222 to assess its subset-specific effect on cytokine production. As with their human counterparts, cytokine production was impaired by ABHD11 inhibition, with a reduction in both the frequency of IFNγ-, IL-13- and IL-17-producing cells, as well as the quantity of each cytokine, under Th1-, Th2- and Th17-polarising conditions, respectively (Fig. 1j–l). These findings were largely recapitulated, with the exception of Th1-mediated IFNγ production, when murine T-cells were treated with ML-226, an inhibitor primarily associated with the human ortholog of ABHD11, but has some affinity for the equivalent murine protein (Supplementary Fig. 1g–i). It should be noted that WWL222 does not bind human ABHD11[24], therefore we did not perform the reciprocal experiments. Assessment of ABHD11 inhibition on murine Treg polarisation did not reveal any defects in FOXP3 expression when cultured with either WWL222 or ML-226 (Supplementary Fig. 1j). Furthermore, no compensatory increases in FOXP3 were observed under any of the three polarising conditions (Th1, Th2 or Th17) upon treatment with either ML-226 or WWL222 (Supplementary Fig. 1k–m). Together, these data indicate that ABHD11 is essential for optimal T-cell function.

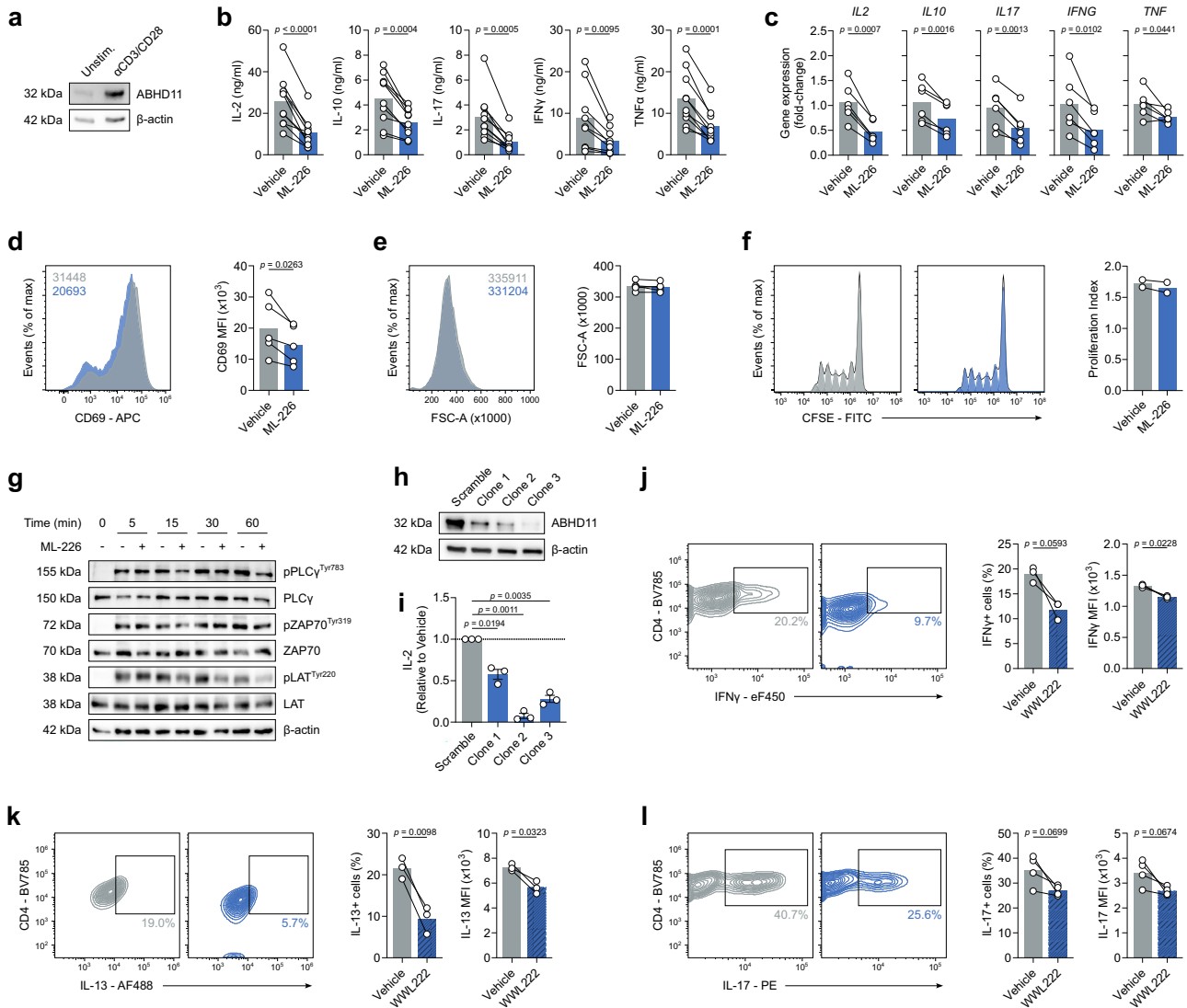

**Fig. 1 | ABHD11 inhibition impairs T-cell activation and cytokine production.**
**a** ABHD11 expression in CD4+ T-cells, unstimulated or activated with α-CD3 and α-CD28 ($n = 3$). Protein loading assessed using β-actin. **b** IL-2, IL-10, IL-17, IFNγ and TNFα production by CD4+ effector T-cells (IL-2, IL-10, IL-17, TNFα: $n = 10$; IFNγ: $n = 9$). For $p < 0.0001$, $p = 0.00007741$. **c** qPCR analysis of *IL2, IL10, IL17, IFNG* and *TNF* expression in CD4+ effector T-cells ($n = 6$). **d** Surface expression of CD69, as a measure of activation, on CD4+ effector T-cells ($n = 5$). **e** Cell size, as determined by forward scatter area, of CD4+ effector T-cells ($n = 5$). **f** Proliferation, as determined by CFSE staining, in CD4+ T-cells at 72 h ($n = 2$). **g** T-cell receptor signalling protein phosphorylation and expression in CD4+ T-cells, unstimulated or activated with α-CD3 and α-CD28 for the time points indicated ($n = 3$). Protein loading assessed using β-actin. **h** ABHD11 expression in unstimulated ABHD11 knockdown Jurkat T-cells ($n = 3$). Protein loading assessed using β-actin. **i** IL-2 production by ABHD11

knockdown Jurkat T-cells ($n = 3$). **j** Intracellular IFNγ expression in murine CD4+ effector T-cells following polarisation towards Th1 in the presence and absence of WWL222 ($n = 3$). **k** Intracellular IL-13 expression in murine CD4+ effector T-cells following polarisation towards Th2 in the presence and absence of WWL222 ($n = 3$). **l** Intracellular IL-17 expression in murine CD4+ effector T-cells following polarisation towards Th17 in the presence and absence of WWL222 ($n = 4$). Experiments were carried out using human samples, unless otherwise stated. CD4+ T-cells were activated with α-CD3 (2 μg/ml) and α-CD28 (20 μg/ml) for 24 h, in the presence and absence of ML-226, unless otherwise stated. Data are expressed as either: mean, with paired dots representing data from distinct biological replicates; or mean ± SEM. Statistical tests used: two-tailed paired *t*-test (**b**–**f**, **j**–**l**), two-tailed one-sample *t*-test (**i**). Source data are provided as a Source Data file.

## ABHD11 inhibition rewires human T-cell metabolism

Given that ABHD11 primarily maintains the function of α-KGDH within the TCA cycle[22], we next investigated the impact of ABHD11 inhibition on human T-cell metabolism. Firstly, we assessed whether ML-226 directly inhibits the catalytic activity of α-KGDH. Although modest, we recorded a significant reduction in α-KGDH activity (Fig. 2a), indicating that ML-226 can inhibit ABHD11 oxidative decarboxylation of α-KG in T-cells. Interestingly, inhibition of α-KGDH using an alternative small molecule inhibitor (CPI-613; currently undergoing clinical evaluation [NCT05325281]) was able to reproduce the reduction in cytokines observed upon ABHD11 inhibition (Supplementary Fig. 2a). Subsequently, we activated T-cells in the presence and absence of ML-226

and examined cellular metabolism using a mitochondrial stress assay. Here, there was a significant reduction in oxygen consumption rate (OCR) following ABHD11 inhibition, with consistent reductions in basal respiration, ATP-linked respiration, maximal respiratory capacity and spare respiratory capacity (Fig. 2b, c). As expected, these changes precede impaired ATP production from OXPHOS (Fig. 2d). Furthermore, we utilised targeted mass spectrometry analysis to understand potential alterations at the metabolite level. Here, ABHD11 inhibition revealed a compromised TCA cycle, in which succinate levels were significantly reduced following ABHD11 inhibition, in addition to a striking accumulation of acetyl-CoA (Fig. 2e). To determine whether the source of acetyl-CoA was citrate-derived, and thus dependent on

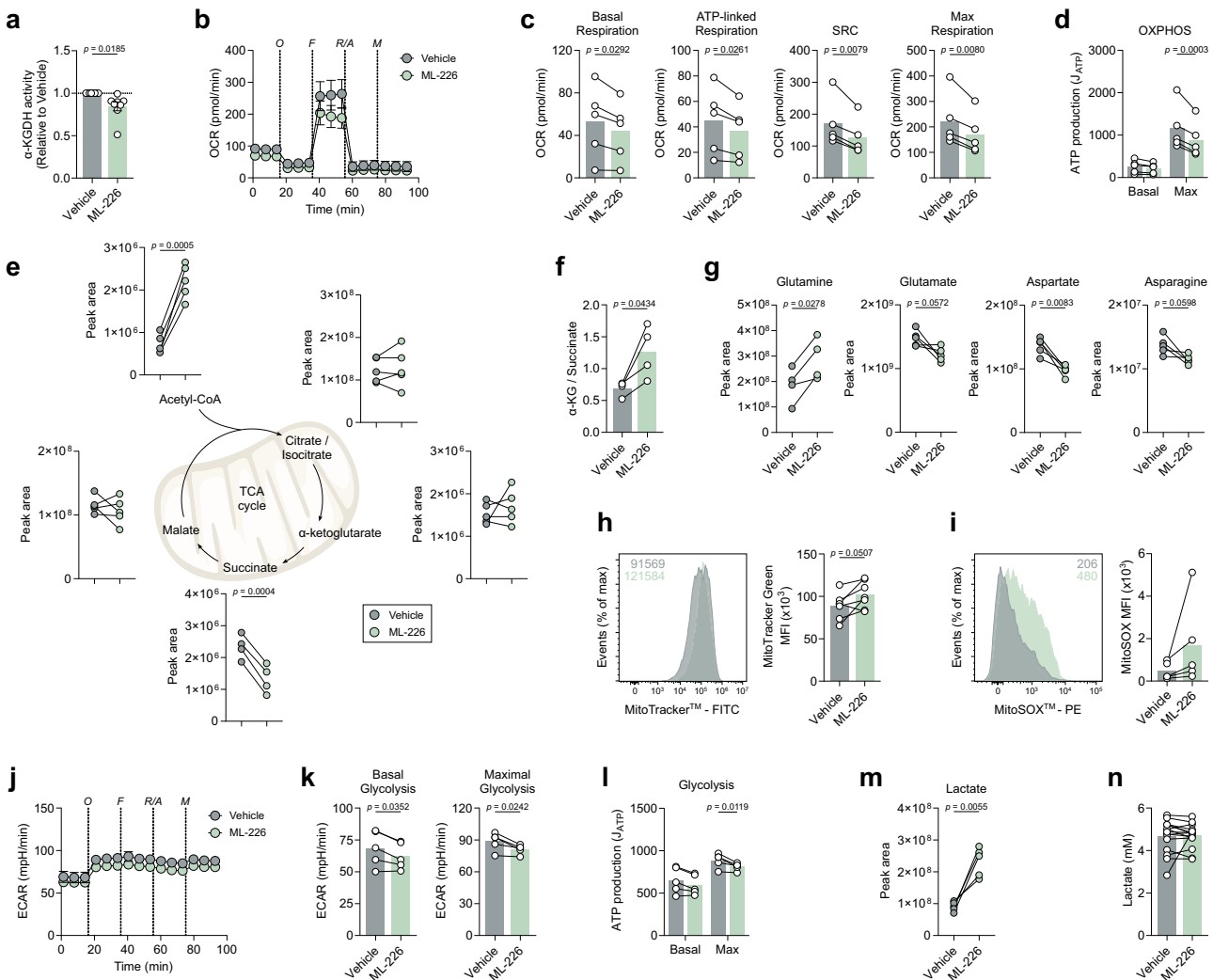

**Fig. 2 | ABHD11 inhibition rewires mitochondrial metabolism in human T-cells.**
**a** α-ketoglutarate dehydrogenase (α-KGDH) activity in CD4+ effector T-cells (n = 8).
**b** Oxygen consumption rate (OCR) in CD4+ effector T-cells (n = 5). Pre-optimised injections include: oligomycin, FCCP, antimycin A/rotenone (all 1 μM) and monensin (20 μM). **c** Calculated OCR parameters from OCR measured in (**b**), including basal respiration, ATP-linked respiration, spare respiratory capacity and maximal respiratory capacity (n = 5). **d** Basal and maximal ATP production ($J_{ATP}$) from OCR measured in (**b**; n = 5). **e** Intracellular levels of selected TCA cycle intermediates in CD4+ effector T-cells. Metabolites include: acetyl-CoA, citrate/isocitrate, α-ketoglutarate (α-KG), succinate and malate (acetyl-CoA, citrate/isocitrate, α-KG, malate: n = 5; succinate: n = 4). **f** Determination of intracellular α-ketoglutarate to succinate ratio in CD4+ effector T-cells (n = 4). **g** Intracellular levels of selected amino acids in CD4+ effector T-cells. Metabolites include: glutamine, glutamate, aspartate and asparagine (glutamine: n = 4; glutamate, aspartate, asparagine: n = 5). **h** Mitochondrial content, as determined by MitoTracker™ Green, in CD4+ effector

T-cells (n = 7). **i** Mitochondrial reactive oxygen species, as determined by Mito-SOX™ Red, in CD4+ T-cells at 1 h (n = 5). **j** Extracellular acidification rate (ECAR) in CD4+ effector T-cells (n = 5). Pre-optimised injections include: oligomycin, FCCP, antimycin A/rotenone (all 1 μM) and monensin (20 μM). **k** Calculated ECAR parameters from ECAR measured in (**j**), including basal glycolysis and maximal glycolysis (n = 5). **l** Basal and maximal ATP production ($J_{ATP}$) from ECAR measured in (**j**; n = 5). **m** Intracellular levels of lactate in CD4+ effector T-cells (n = 5). **n** Extracellular lactate in cell-free supernatants from cultures of CD4+ effector T-cells (n = 15). All experiments were carried out using human samples. CD4+ T-cells were activated with α-CD3 (2 μg/ml) and α-CD28 (20 μg/ml) for 24 h, in the presence and absence of ML-226, unless otherwise stated. Data are expressed as either: mean, with paired dots representing data from distinct biological replicates; or mean ± SEM. Statistical tests used: two-tailed one-sample t-test (**a**), two-tailed paired t-test (**c**, **e**–**i**, **k**, **m**, **n**), two-way ANOVA with Šidák's multiple comparisons test (**d**, **l**). Source data are provided as a Source Data file.

ATP citrate lyase (ACLY), we cultured T-cells in the presence and absence of a specific ACLY inhibitor, BMS-303141. To this end, we did not observe a rescue in cytokine production upon culture with BMS-303141 (Supplementary Fig. 2b), suggesting that this conversion of mitochondrial citrate to cytosolic acetyl-CoA is dispensable. An alternative mechanism of buffering excess mitochondrial acetyl-CoA, independent of citrate, is conversion to acetyl-carnitine[26]. Analysis of our mass spectrometry data confirmed heightened levels of acetyl-carnitine following ABHD11 inhibition (Supplementary Fig. 2c), which together suggests that the carnitine shuttle buffers excess mitochondrial acetyl-CoA upon ABHD11 inhibition. Although α-KG levels remain similar following ABHD11 inhibition (Fig. 2e), there is a marked increase

in the observed α-KG / succinate ratio within these cells (Fig. 2f), further indicating impaired α-KGDH activity. However, intracellular 2-HG levels were unchanged (Supplementary Fig. 2d), suggesting that there is alternative mechanism underpinning the observed phenotype, rather than the epigenetic mechanism previously described following ABHD11 loss[22]. Moreover, the amino acid pool is also noticeably diminished in these cells—with the exception of glutamine, whereby increased levels are reported—with observed reductions in glutamate, aspartate and asparagine (Fig. 2g). Aspartate plays an important role in nucleotide synthesis[27]; therefore, it is unsurprising that there is a concomitant reduction in the production of several nucleotides following ABHD11 inhibition (Supplementary Fig. 2e). Interestingly,

T-cells treated with ML-226 appeared to have greater mitochondrial mass (Fig. 2h), though this translated into no discernible effect on mitochondrial depolarisation (Supplementary Fig. 2f). Blockade through α-KGDH is associated with the production of mitochondrial reactive oxygen species (mitoROS)[28], therefore we assessed both mitoROS and total ROS levels using mitoSOX and cellROX, respectively. Here, mitoROS levels were initially heightened 1 h post-ABHD11 inhibition, but this difference later dissipated at 24 h (Fig. 2i and Supplementary Fig. 2g). Total ROS levels were elevated at 1 and 24 h (Supplementary Fig. 2h), in agreement with the observed reduction in the glutathione pool present in these cells (Supplementary Fig. 2i). Taken together, there is substantial evidence that the TCA cycle is compromised following ABHD11 inhibition, altering the metabolic landscape within the mitochondria.

Given the impact on OXPHOS, we next determined whether ABHD11 inhibition perturbed lactic acid excretion, measured using the extracellular acidification rate (ECAR). We observed a modest reduction in ECAR, whereby glycolytic capacity was attenuated by ABHD11 inhibition (Fig. 2j, k). This translated into a reduction in glycolytic ATP production (Fig. 2l). However, these changes are not necessarily underpinned by depletion of the metabolite pool, as intracellular levels of glycolytic intermediates are not consistently changed following ABHD11 inhibition (Supplementary Fig. 2j). A substantial increase of intracellular lactate levels was observed upon ABHD11 inhibition (Fig. 2m). Importantly, lactate accumulates within these cells and is not exported at a higher rate (Fig. 2n), which agrees with the modest reduction in extracellular acidification rate with ABHD11 inhibition (Fig. 2j–l). Together, these data show that ABHD11 inhibition rewires mitochondrial metabolism, establishing a compromised TCA cycle that promotes the accumulation of acetyl-CoA and lactate.

To determine the source of the elevated lactate, we next employed liquid chromatography mass spectrometry to track the fate of cellular metabolites. Here, we performed stable isotope labelling using uniformly-labelled $^{13}C_6$-glucose to follow the incorporation of these carbons into downstream metabolites. An elevated percentage of $^{13}C$ is incorporated into lactate (Supplementary Fig. 3a), which perhaps suggests a feedback mechanism from the compromised TCA cycle. We also observed $^{13}C$ incorporation into acetyl-CoA. Strikingly, $^{13}C$ incorporation into succinate is unchanged and there was relatively little incorporation of glucose carbon downstream of α-KG, perhaps due to timepoint restrictions. Therefore, we utilised uniformly-labelled $^{13}C_5$-glutamine (almost directly upstream of α-KGDH) to measure glutamine anaplerosis (Supplementary Fig. 3b). Here, reduced $^{13}C$ incorporation into succinate was more pronounced, whilst incorporation into other TCA cycle intermediates did not appear to be significantly altered (Supplementary Fig. 3b). Whilst we did not observe a build-up of $m+5$ α-KG, inhibition of α-KGDH is reflected by a significant reduction in $m+4$ succinate and an increased ratio of α-KG$^{m+5}$/succinate$^{m+4}$ (Supplementary Fig. 3c–e). These data would suggest that the accumulation of lactate and most likely acetyl-CoA is sustained by glucose following ABHD11 inhibition.

**ABHD11 inhibition upregulates SREBP signalling to drive oxysterol synthesis and activate LXR**

To further our understanding of the mechanisms underpinning rewired T-cell metabolism and suppressed effector function following ABHD11 inhibition, we sought to investigate changes in the T-cell transcriptome following ML-226 treatment. RNA-Seq analysis revealed 69 genes that were differentially-expressed upon ABHD11 inhibition, of which 38 were upregulated and 31 were downregulated (Fig. 3a). To realise the biological relevance of these changes, we performed pathway enrichment analysis to determine which pathways become up- and downregulated upon ABHD11 inhibition. Unsurprisingly, several pathways associated with T-cell function, such as *cellular response to cytokine stimulus* and *inflammatory response*, were downregulated

following ABHD11 inhibition (Fig. 3b), supporting our observed effects on cytokine production (Fig. 1b, c). Conversely, the vast majority of the pathways upregulated in response to ABHD11 inhibition were associated with either sterol or fatty acid metabolism, with *sterol biosynthetic process* emerging as the most enriched pathway overall (Fig. 3b). Interestingly, RNA-Seq analysis of ABHD11 knockdown Jurkat T-cell clones highlighted the upregulation of similar metabolic pathways (sterol and cholesterol biosynthetic processes; Supplementary Fig. 4a). These data would suggest that the immunomodulatory phenotype observed are likely attributed to sterol biosynthetic processes, rather than alternative acetyl-CoA-mediated processes.

To determine what drives the observed changes in gene expression and, ultimately, the suppressed effector function observed following ABHD11 inhibition, we performed transcription factor analysis. Here, enriched transcription factors are predicted using the differentially-expressed genes identified—the lower the "average rank" score, the more enriched that transcription factor and its activity. Once more, there was a clear association with lipid metabolism, whereby *PPARG*, *SREBF1* and *SREBF2* were all amongst the most enriched transcription factors (Fig. 3c). In fact, *SREBF1* and *SREBF2* are significantly upregulated at the transcript level following ABHD11 inhibition (Fig. 3a). In further support of this, *sterol biosynthetic process* is the pathway most significantly enriched following ABHD11 inhibition (Fig. 3d)—a process tightly regulated by the signalling of sterol regulatory element binding proteins (SREBPs) encoded by *SREBF1* and *SREBF2*[29]. There is recent evidence demonstrating the existence of a lactate-SREBP2 signalling axis within human immune cells[30]. Therefore, given the heightened intracellular lactate levels we observe following ABHD11 inhibition (Fig. 2m), we activated T-cells in the presence of either ML-226 or lactic acid and assessed whether there was a comparable effect on SREBF2 expression and cytokine production. Here, treatment with lactic acid phenocopied ABHD11 inhibition, with a similar increase in *SREBF2* mRNA levels accompanied by a corresponding reduction in IL-2 and IFNγ production (Supplementary Fig. 4b, c). These data suggest that lactate drives SREBP activation and the subsequent upregulation of sterol biosynthesis pathways.

To establish the significance of augmented sterol biosynthesis following ABHD11 inhibition, we next carried out specialised mass spectrometry to measure the intracellular levels of various sterol species. Surprisingly, we observed a reduction in total sterol levels following ABHD11 inhibition (Fig. 3e). This is primarily attributed to a reduction in non-oxygenated sterols, wherein there is a trend towards decrease across the group, with several significantly reduced (Fig. 3f; Supplementary Fig. 4d). Conversely, we identified heightened oxysterol levels within these cells (Fig. 3g), which in turn increased the oxysterol/sterol ratio (Supplementary Fig. 4e). Of the oxysterols measured, 27-hydroxycholesterol (27-HC) and 24,25-epoxycholesterol (24,25-EC) emerged as the two most significantly elevated species, particularly 24,25-EC whose levels are approximately 5–10 times higher following ABHD11 inhibition (Fig. 3h). Interestingly, there does not appear to be a general increase across all oxysterol species, as 7α-hydroxycholesterol (7α-HC) and 24-hydroxycholesterol (24-HC) levels were reduced upon ABHD11 inhibition (Fig. 3h). Intriguingly, we report activation of a shunt pathway—branching from the classical mevalonate pathway at oxidosqualene—which synthesises 24,25-EC from acetyl-CoA (Supplementary Fig. 4f).

27-HC and 24,25-EC are potent activators of liver X receptor (LXR) signalling[31]. To establish whether the increased production of 27-HC and 24,25-EC following ABHD11 inhibition leads to LXR activation, we compared treatment with ML-226 versus treatment with GW3965—a synthetic LXR agonist[32]. Interestingly, recent work by Waddington et al. assessed the transcriptional effects of LXR activation on human CD4+ T-cell function using RNA-Seq[33]. In comparison to our dataset, 19 of the 65 LXR-regulated transcripts identified were also differentially regulated by ABHD11 inhibition (Fig. 3i), which accounts for approximately

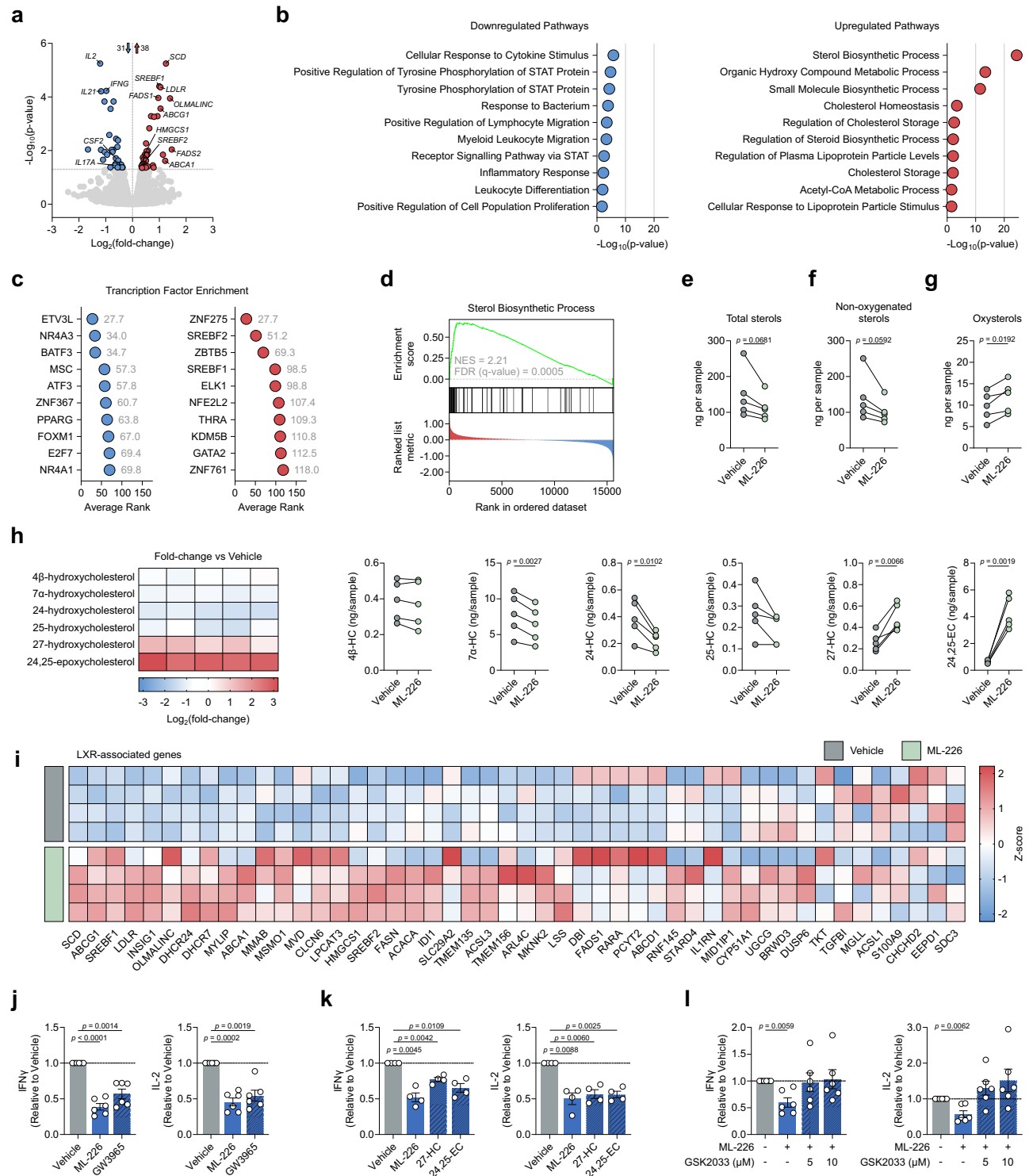

27.5% of all differentially-expressed transcripts (Supplementary Fig. 4g), indicating significant overlap between both nodes. As such, we hypothesised that LXR activation would phenocopy, at least partially, ABHD11 inhibition in human T-cells. To this end, we treated T-cells with ML-226 and GW3965 in parallel before assessing cytokine production. Here, we observed comparable reductions in IFNγ and IL-2 production between ABHD11 inhibition and LXR activation (Fig. 3j). This effect was also observed when treating T-cells with the intracellular oxysterol species that are increased following ABHD11 inhibition—27-HC and 24,25-EC (Fig. 3k). Interestingly, co-treatment with the LXR antagonist, GSK2033, ablated the mRNA levels of LXR targets SREBF1 and SCD

following ABHD11 inhibition (Supplementary Fig. 4h), and subsequently rescued the impaired cytokine response (Fig. 3l). Together, these data suggest that LXR activation drives, at least in part, the anti-inflammatory phenotype observed in human T-cells following ABHD11 inhibition.

Given that acetyl-CoA both accumulates within the cell (Fig. 2e) and fuels many of the pathways that become upregulated following ABHD11 inhibition (Fig. 3b), we next explored some of the multi-faceted roles of acetyl-CoA. Firstly, we observed no difference in any of the histone acetylation marks measured or total protein acetylation following ABHD11 inhibition (Supplementary Fig. 5a, b). Intriguingly, lipidomic analysis demonstrated markedly increased levels of

**Fig. 3 | ABHD11 inhibition activates SREBP signalling to drive oxysterol synthesis. a** Differential expression analysis by RNA-Seq in CD4+ effector T-cells (*n* = 4). Blue and red data points represent downregulated and upregulated genes, respectively. Transcripts with an adjusted *p*-value < 0.05 were considered differentially expressed. **b** Pathway enrichment analysis based on differentially-expressed genes (*n* = 4). Top 10 enriched pathways are shown. **c** Transcription factor enrichment analysis based on differentially-expressed genes (*n* = 4). The lower the "Average Rank" value, the more enriched the transcription factor activity. Top 10 enriched transcription factors are shown. **d** GSEA enrichment plot for *Sterol Biosynthetic Process* (*n* = 4). **e** Total intracellular sterol levels in CD4+ T-cells (*n* = 5). **f** Intracellular non-oxygenated sterol levels in CD4+ T-cells (*n* = 5). **g** Intracellular oxysterol levels in CD4+ T-cells (*n* = 5). **h** Intracellular levels of selected oxysterols in CD4+ T-cells. Metabolites include: 4β-hydroxycholesterol, 7α-hydroxycholesterol, 24-hydroxycholesterol, 25-hydroxycholesterol, 27-hydroxycholesterol, 24,25-epoxycholesterol (*n* = 5). Heatmap represented as Log$_2$(fold-change) versus vehicle

control. **i** Expression of LXR-associated genes in CD4+ T-cells (*n* = 4). Heatmap represented as individual gene *Z*-scores. **j** IFNγ and IL-2 production by CD4+ T-cells, activated in the presence and absence of ML-226 or GW3965 (*n* = 6). For *p* < 0.0001, *p* = 0.00004089. **k** IFNγ and IL-2 production by CD4+ T-cells, activated in the presence and absence of ML-226 or oxysterols 27-hydroxycholesterol and 24,25-epoxycholesterol (*n* = 4). **l** IFNγ and IL-2 production by CD4+ T-cells, activated in the presence and absence of ML-226 and GSK2033 (*n* = 6). All experiments were carried out using human samples. CD4+ T-cells were activated with α-CD3 (2 μg/ml) and α-CD28 (20 μg/ml) for 24 h, in the presence and absence of ML-226, unless otherwise stated. Data are expressed as either: mean, with paired dots representing data from distinct biological replicates; or mean ± SEM. Statistical tests used: differential expression analysis via eBayesian fit of trimmed mean of *M*-values (**a**), two-tailed paired *t*-test (**e**–**h**), two-tailed one-sample *t*-test (**j**–**l**). Source data are provided as a Source Data file.

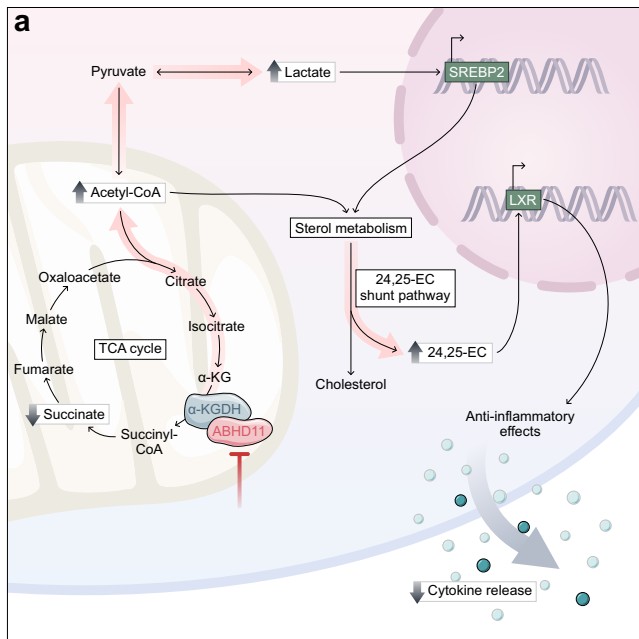

**Fig. 4 | ABHD11 inhibition drives sterol metabolism to modulate T-cell effector functions. a** Schematic overview of the mechanism through which ABHD11 inhibition suppresses T-cell effector function. ABHD11 activity is required for the conversion of α-ketoglutarate (α-KG) to succinyl-CoA. The TCA cycle becomes compromised following ABHD11 inhibition, resulting in increased levels of acetyl-CoA and lactate. Increased lactate drives SREBP2 activation, which together with increased acetyl-CoA levels promotes sterol metabolism, particularly the generation of 24,25-epoxycholesterol (24,25-EC) via the shunt pathway. Liver X receptor (LXR) activation increases in response to increased levels of 24,25-EC, which ultimately suppresses T-cell effector functions including cytokine production.

triacylglycerols (TAGs) in response to ABHD11 inhibition (Supplementary Fig. 5c–e), accompanied by a trend towards increase in diacylglycerol (DAG) levels (Supplementary Fig. 5c, d). These findings are in agreement with a previous study by Waddington et al., wherein pharmacological activation of LXR within human T-cells was shown to significantly heighten the levels of several TAG species[33]. Overall, these data indicate that surplus acetyl-CoA within ABHD11-inhibted T-cells is used to fuel the lipid-associated processes that are upregulated by LXR activation, generating increased levels of TAGs.

### ABHD11 inhibition suppresses T-cell effector function in human rheumatoid arthritis and type 1 diabetes

Our findings thus far have outlined that ABHD11 inhibition suppresses T-cell effector function, underpinned by a compromised TCA cycle that promotes 24,25-EC synthesis and consequently

activates LXR signalling (Fig. 4). This highlights the exciting potential of manipulating ABHD11 function for therapeutic benefit in T-cell-mediated autoimmune disease, where reversing the hyper-activation and -function of pathogenic T-cells would be valuable. We explored this possibility initially in two autoimmune patient cohorts, isolating CD4+ T-cells from rheumatoid arthritis (RA) and type 1 diabetes (T1D) patients and activating them in the presence and absence of ML-226 ex vivo (Fig. 5a). Here, we again observed a significant reduction in the production of a diverse range of cytokines by RA and T1D T-cells following ABHD11 inhibition (Fig. 5b, c), alongside a modest, but significant reduction in T-cell activation (Fig. 5d, e). Again, there was a minimal reduction in cell size in both patient cohorts (Supplementary Fig. 6a, b), whilst impaired effector function was independent of any changes in viability (Supplementary Fig. 6c, d).

To consolidate these findings, we assessed the efficacy of ABHD11 inhibition on synovial fluid mononuclear cells (SFMCs) isolated from the site of inflammation in a cohort of RA patients (Fig. 5f). To this end, we observed targeted reductions in IL-17 and IFNγ production following ABHD11 inhibition, whilst IL-2, IL-10 and TNFα production remained unchanged (Fig. 5g). In line with our earlier activation data (Fig. 1d), CD4+ T-cell activation remained intact following ABHD11 inhibition (Fig. 5h), with a modest, but significant increase in cell size (Supplementary Fig. 6e). Crucially, these changes did not result from compromised viability (Supplementary Fig. 6f). Together, these findings demonstrate that ABHD11 inhibition retains its suppressive effect on T-cell function in patients with RA and T1D, including those present at the site of inflammation.

### ABHD11 inhibition delays the onset of murine type 1 diabetes

To support our findings in humans, we next investigated whether ABHD11 regulates T-cell fate and function in a murine model of accelerated T1D. Here, we initially assessed the effect of ABHD11 inhibition on antigen-specific CD4+ T-cells, whereby diabetogenic H2-Ag7-restricted BDC2.5 CD4+ T-cells were stimulated in vitro in the presence of their cognate antigen[34] (a hybrid insulin peptide [HIP]) and treated with the murine ABHD11 inhibitor, WWL222. There was a minimal reduction in proliferation observed following ABHD11 inhibition (Fig. 6a). Despite a modest, but significant reduction in CD25 and CD69 expression on proliferating cells, the proportion of cells expressing these markers was unchanged (Fig. 6b), again indicating that their activation is intact. Importantly, proinflammatory cytokine production is impaired by ABHD11 inhibition, with marked reductions in IFNγ, IL-2, IL-17 and TNFα (Fig. 6c). Interestingly, this suppression appeared to be limited to proinflammatory cytokines, as we observed a striking increase in IL-10 production following ABHD11 inhibition (Fig. 6c), which might indicate that ABHD11 inhibition induces an anti-inflammatory phenotype in antigen-specific T-cells, rather than generally inhibiting effector function.

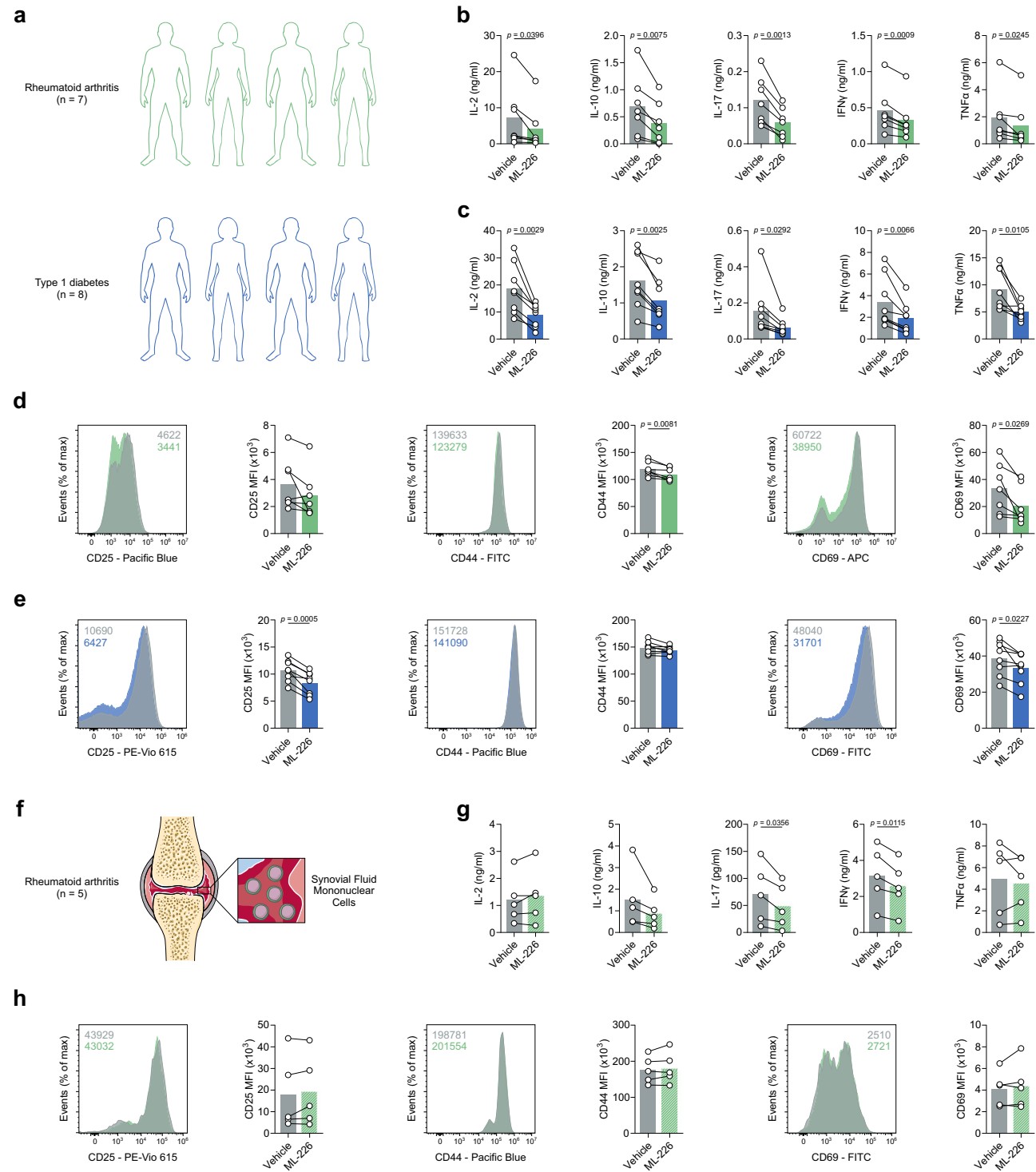

**Fig. 5 | ABHD11 inhibition impairs CD4+ T-cell function in autoimmunity.**
**a** Experimental design of rheumatoid arthritis (RA) and type 1 diabetes (T1D) patient cohorts. **b**, **c** IL-2, IL-10, IL-17, IFNγ and TNFα production by patient-derived CD4+ T-cells from (**b**) RA ($n = 7$) and **c** T1D ($n = 8$) cohorts. **d**, **e** Surface expression of activation markers (CD25, CD44 and CD69) on patient-derived CD4 + T-cells from (**d**) RA ($n = 7$) and **e** T1D ($n = 8$) cohorts. **f** Experimental design of RA patient synovial fluid mononuclear cells (SFMCs). **g** IL-2, IL-10, IL-17, IFNγ and TNFα production by patient-derived SFMCs ($n = 5$). (**h**) Surface expression of activation markers (CD25, CD44 and CD69) on patient-derived SFMCs ($n = 5$). All experiments were carried out using human samples. CD4+ T-cells were activated with α-CD3 (2 µg/ml) and α-CD28 (20 µg/ml) for 24 h, in the presence and absence of ML-226, unless otherwise stated. Data are expressed as mean, with paired dots representing data from distinct biological replicates. Statistical tests used: two-tailed paired *t*-test (**b**–**e**, **g**, **h**). Source data are provided as a Source Data file.

We furthered these investigations using an in vivo murine model of accelerated T1D. To this end, immunocompromised Rag-deficient female mice were injected with $5 \times 10^6$ peptide-activated BDC2.5 CD4+ T-cells and WWL222 (or vehicle control), with the drug dose repeated daily for the course of the experiment (Fig. 7a). It is well-documented

that the Rag1$^{-/-}$ adoptive transfer model presents a rapid and high-penetrance disease course[35–37], thereby small delays in onset can reflect meaningful biological activity. Encouragingly, ABHD11 inhibition by low-dose WWL222 treatment significantly delayed the development of T1D (Fig. 7b), attributable to their glycaemic stability versus the vehicle

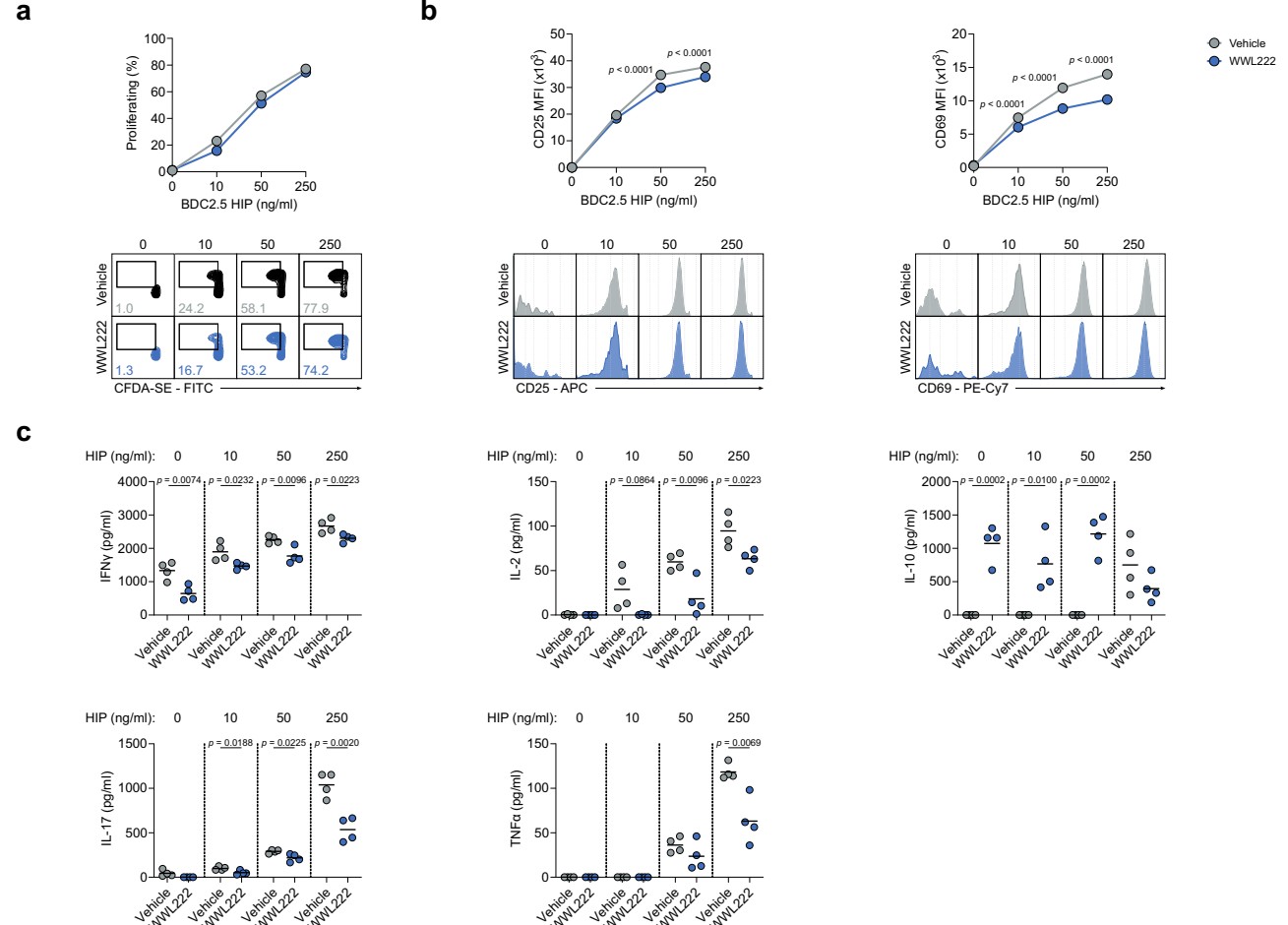

**Fig. 6 | ABHD11 inhibition impairs the function of antigen-specific T-cells.**
**a** Frequency of proliferating cells, as determined using CFDA-SE, on antigen-specific T-cells ($n = 4$). **b** Surface expression of activation markers (CD25 and CD69) on antigen-specific T-cells ($n = 4$). **c** IFNγ, IL-2, IL-10, IL-17 and TNFα production by antigen-specific T-cells ($n = 4$). All experiments were carried out using murine samples. BDC2.5 CD4+ T-cells were activated with hybrid insulin peptides (HIPs), in the presence and absence of WWL222. Data are expressed as mean. Statistical tests used: two-way ANOVA with Šidák's multiple comparisons test (**a**, **b**), multiple two-tailed unpaired *t*-test (**c**). Source data are provided as a Source Data file.

control group (Supplementary Fig. 7a). At the end of the observation period, immune cells were harvested from the spleen of diabetic mice to elucidate the changes that underpin delayed onset of disease. Here, CD4+ T-cells from WWL222-treated mice displayed impaired activation (Fig. 7c), and produced significantly less IFNγ, TNFα and IL-2 (Fig. 7d; Supplementary Fig. 7b, c) versus those harvested from untreated mice. Additionally, TNFα was reduced in myeloid cell populations, meaning ABHD11 inhibition may skew the cytokine profile through multiple cell types (Supplementary Fig. 7d–f). To strengthen and validate these findings, we utilised an additional murine model in which immunocompromised Rag-deficient female mice were injected with $12 \times 10^6$ splenocytes from newly diabetic mice and treated as previously described with WWL222 (Fig. 7e). Excitingly, ABHD11 inhibition again significantly delayed the onset of T1D (Fig. 7f) by maintaining blood glucose levels (Supplementary Fig. 7g). Assessment of splenocytes harvested post-observation period revealed that this alleviation of T1D was underpinned by a reduction in the frequency of IFNγ-producing CD4+ T-cells (Fig. 7g), with no change in TNFα production (Supplementary Fig. 7h). These data show that targeting ABHD11 can improve outcomes in settings of T-cell-mediated inflammation, highlighting the significant potential of developing drugs that inhibit this metabolic node as a novel treatment strategy.

## Discussion

Reduced expression of ABHD11, a mitochondrial serine hydrolase, within patient CD4+ T-cells is associated with clinical remission in RA[21]. However, beyond its role in maintaining the function of α-KGDH[22], the function of ABHD11 in human T-cells remains unknown. In this paper, we show that effector cytokine production is impaired following ABHD11 inhibition, underpinned by the rewiring of mitochondrial metabolism. We thereby present ABHD11 as a potential drug target, through which to restrict T-cell effector function.

A study from Bailey et al., first described that ABHD11 maintains functional lipoylation of the DLST subunit of α-KGDH[22], which catalyses the conversion of αKG to succinyl-CoA within the TCA cycle. Importantly, we demonstrate that α-KGDH activity is impaired in T-cells treated with ML-226, a highly selective ABHD11 inhibitor, where there is a reduction in cellular succinate and a corresponding increase in the α-KG to succinate ratio. Accumulation of α-KG and subsequent formation of 2-HG can impair the activity of α-KG-dependent dioxygenases[38–41], which play an important role in epigenetic processes such as histone modifications, amongst other activities[42]. However, intracellular 2-HG levels were unchanged in T-cells upon ABHD11 inhibition. Instead, the compromised TCA cycle drives the accumulation of lactate and acetyl-CoA. Despite evidence that high concentrations of acetyl-CoA enhance histone acetylation to

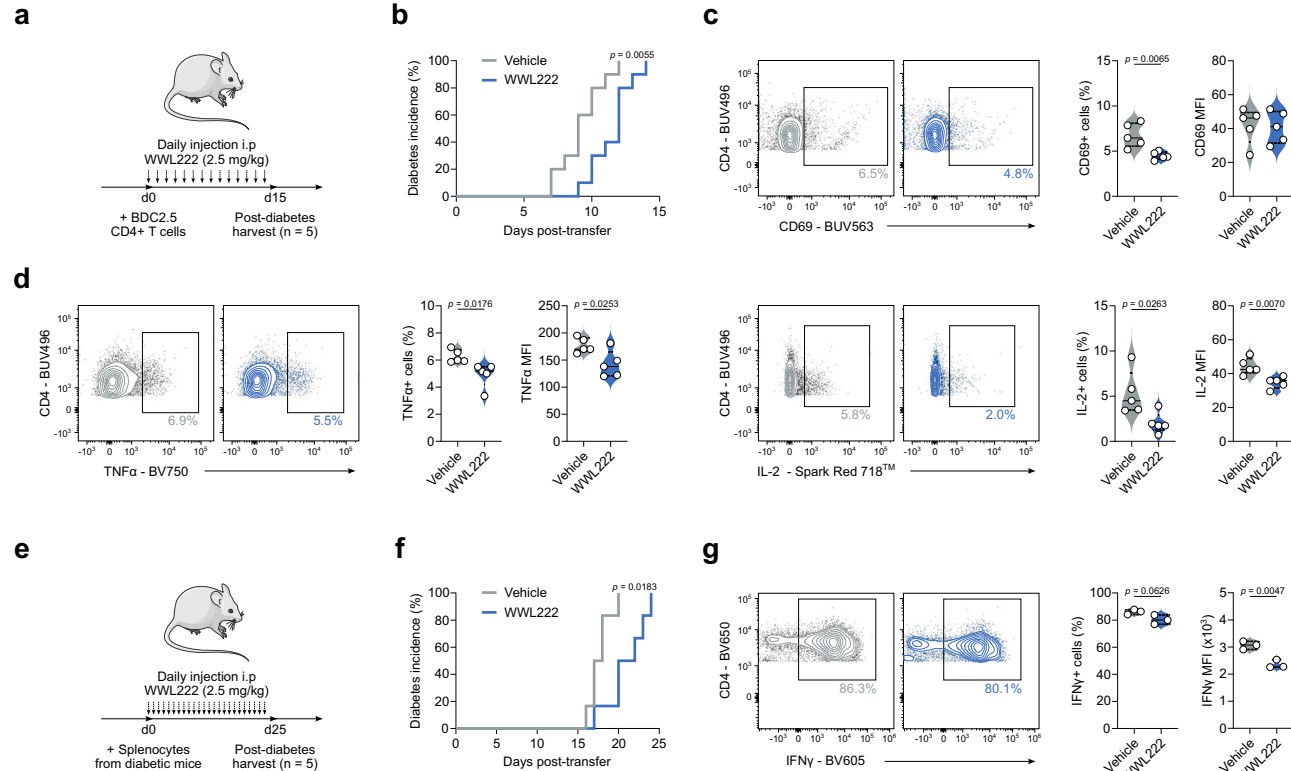

**Fig. 7 | ABHD11 inhibition delays the onset of type 1 diabetes. a** Schematic overview of in vivo diabetes adoptive transfer model using BDC2.5 HIP-activated BDC2.5 CD4+ T-cells. **b** Diabetes incidence, as confirmed by blood glucose measurement >13.9 mmol/L, in the presence and absence of daily injections i.p of 2.5 mg/kg WWL222 (*n* = 10). **c** Surface expression of CD69, as a measure of activation, on CD4+ T-cells (*n* = 5). **d** Intracellular TNFα and IL-2 expression in CD4+ T-cells (*n* = 5). **e** Schematic overview of in vivo diabetes adoptive transfer model using splenocytes. **f** Diabetes incidence, as confirmed by blood glucose measurement >13.9 mmol/L, in the presence and absence of daily injections i.p of 2.5 mg/kg WWL222 (*n* = 6). **g** Intracellular IFNγ expression in CD4+ T-cells (*n* = 3). All experiments were carried out using murine samples. Mice were injected daily with the indicated dose of WWL222. Data are expressed as median ± interquartile range. Statistical tests used: log-rank (Mantel–Cox) test (**b**, **f**), two-tailed unpaired *t*-test (**c**, **d**, **g**). Source data are provided as a Source Data file.

subsequently shape T-cell responses[43], we did not observe any significant changes in histone acetylation following ABHD11 inhibition.

We instead reveal that ABHD11 inhibition restricts human T-cell function through a SREBP2-mediated increase in oxysterol synthesis. SREBPs are considered to be the master regulators of lipid and sterol biosynthesis[44]. Although SREBP activity is required for the metabolic programming that primes T-cells for growth and proliferation[45,46], the impact of augmented SREBP activation has yet to be fully explored in T-cells. A recent study has described a lactate-SREBP2 signalling axis, whereby dendritic cells exposed to increased levels of lactate within the tumour microenvironment promoted cholesterol metabolism through a SREBP2-dependent pathway[30]. This appears to be driven by the concomitant change in extracellular pH, as an acidic pH (pH 6.8) has previously been shown to induce activation and nuclear translocation of SREBP2, leading to the expression of cholesterol biosynthesis genes[47]. Importantly, this relationship is not restricted to extracellular lactate levels, as the accumulation of lactate within cells has also been shown to promote SREBP2-mediated cholesterol biosynthesis[48]. In line with these studies, we show that following ABHD11 inhibition, there is an accumulation of intracellular lactate, with a concurrent upregulation of SREBP and sterol biosynthesis genes. Notably, this does not appear to be specific to the traditional mevalonate pathway in T-cells, rather we show increased flux through a shunt pathway that produces 24,25-EC.

24,25-EC is one of the most efficacious of the oxysterols in the activation of LXR[31], and we reveal that 24,25-EC levels are approximately 5–10× greater following ABHD11 inhibition. In keeping with a previous report that investigated the role of LXR activation in human

CD4+ T-cell function[33], a significant proportion of the transcripts upregulated upon ABHD11 inhibition overlapped with those upregulated following exposure to a highly-selective LXR agonist GW3965. It is also important to note that LXR activation is closely integrated with SREBP signalling within inflammatory response programmes[49]. Typically, LXR activation is considered to be anti-inflammatory in a host of immune cell types[33,50,51]. Specifically, CD4+ T-cells treated with LXR agonists display reduced proinflammatory function, including a reduction in their IL-17 production[33]. We show comparable reductions in T-cell function following ABHD11 inhibition, treatment with a synthetic LXR agonist, or culture with oxysterols (27-HC and 24,25-EC). Moreover, inhibition of LXR can rescue the impaired cytokine production of ABHD11-inhibited T-cells. 24,25-EC itself has also been shown to suppress immune cell function, limiting iNOS activation in LPS-activated monocytes in an LXR-dependent manner[52]. Thus heightened 24,25-EC levels, and downstream LXR activation, following ABHD11 inhibition culminate in suppressed effector function.

The relationship between metabolism and T-cell effector function has been extensively explored during the past two decades[53,54]. Indeed, aberrant metabolic programmes underpin the pathogenesis of several inflammatory conditions, including autoimmune disease, wherein unregulated T-cell metabolism fuels a hyper-inflammatory phenotype[3,4]. Given the relative ineffectiveness of current treatments[13,55], as well as the debilitating side effects that are often associated with these drugs[14], novel approaches to resolve pathogenic tissue inflammation in autoimmune disease are highly attractive. Multiple pre-clinical studies have explored the prospect of modulating T-cell metabolism to ameliorate inflammation and, thus, improve

disease outcomes—for example, PKM2 tetramerisation[15], suppressing OXPHOS using oligomycin[16], and combined inhibition of glucose metabolism by 2-DG and metformin[17]. Despite these recent successes, the potential toxicity of targeting cellular metabolism at a systemic level remains an important consideration. In this paper, we targeted the typical immunometabolic profile of T-cells by inhibiting ABHD11, with the aim of uncovering a druggable target that has wide applicability. We acknowledge that ABHD11 inhibition might also affect Treg cell populations, as our in vitro polarisation data show a reduction in FOXP3+ T-cells under these conditions, therefore follow-up studies using in vivo Treg reporter models—such as the NOD/FOXP3$^{GFP}$ mouse —would further our understanding of Treg cell dynamics and stability following ABHD11 inhibition. However, whilst ABHD11 inhibition is not truly selective for autoreactive T-cells, our data support that it restricts TCR-dependent activation and downstream cytokine production, which are hallmarks of autoreactive responses.

Crucially, ABHD11 inhibition retained its suppressive effect on T-cell function in two human cohorts of autoimmunity. Firstly, there was a comprehensive reduction in cytokine output when CD4+ T-cells from RA or T1D patients were cultured ex vivo with ML-226. These findings were consolidated using a murine model of accelerated T1D, whereby ABHD11 inhibition suppressed antigen-specific T-cell responses. In contrast to our initial findings, our data suggested that antigen-specific T-cells also produce elevated IL-10 following ABHD11 inhibition. However, it is important to consider that BDC2.5 TCR transgenic NOD mice, wherein our antigen-specific CD4+ T-cells were harvested from, do not develop autoimmune diabetes due to an expanded Treg population[36]. Thus, is it possible that due to the altered proportion of Treg cells in this mouse, there is a difference in how ABHD11 impacts IL-10 production.

Moreover, daily administration of WWL222 delayed the onset of diabetes within an exceptionally aggressive Rag1$^{-/-}$ adoptive transfer model. In this context, even modest delays in onset—as we observed— can indicate tangible therapeutic benefit. For example, treating Treg cells with anti-CD3/CD20, before co-transfer with splenocytes from diabetic mice into NOD/SCID mice (a similar immunodeficient model to the Rag1$^{-/-}$ transfer model), delays the onset of diabetes by approximately 6 days[56]. In humans, anti-CD3 (Teplizumab) and anti-CD20 (Rituximab) treatments have separately been shown to protect and preserve islet β-cell function for upwards of 6 months to 2 years[57–61], thus the delays in diabetes development that we observe in these accelerated models have the potential to translate into longer protection in humans. Importantly, the observed delay in diabetes onset is underpinned by a T-cell-specific reduction in effector function, with only a minor effect on myeloid cell responses. It is important to note that the material harvested post-sacrifice in our in vivo murine models is insufficient to further confirm our mechanistic findings in this context. We acknowledge this limitation and recognise that further work should explore additional in vivo models of autoimmunity that permit further mechanistic and therapeutic investigation. In particular, we recognise that assessing Treg cell function in vivo is important in further understanding the therapeutic potential of ABHD11 inhibition, and is therefore a limitation of this current study.

Collectively, our study demonstrates that ABHD11 is a critical regulator of TCR signalling across T-cell subsets. Whilst ABHD11 inhibition does not selectively target autoreactive T-cells—similarly to other clinically available agents[62]—the broad suppression of T-cell activation is sufficient to delay diabetes development in a preclinical model of autoimmunity. These findings support the immunomodulatory potential of targeting ABHD11 activity in T-cell biology.

## Methods

### Human samples
**Healthy controls.** Peripheral blood was collected from healthy, non-fasted individuals. Peripheral blood mononuclear cells (PBMCs) were isolated via density gradient centrifugation using Lymphoprep™ (STEMCELL Technologies). No sex-linked differences were observed between healthy control male and female samples.

**Autoimmune patient cohorts.** PBMCs isolated from patients with RA or T1D were cryopreserved until use. SFMCs isolated from RA patients during arthroscopic knee surgery were cryopreserved until use. Cohort demographics can be found in Supplementary Tables 1 and 2.

### Human CD4+ T-cell isolation and culture
Human CD4+ T-cells (130-096-533), CD4+ naïve T (Tnv) cells (130-094-131) and CD4+ T effector (Teff) cells (130-094-125) were isolated by magnetic separation using the autoMACS® Pro Separator as per the manufacturer's instructions (Miltenyi). T-cells were activated with plate-bound anti-CD3 (2 µg/ml; OKT3; BioLegend) and soluble anti-CD28 (20 µg/ml; CD28.2; BioLegend) in human plasma-like medium (HPLM; Gibco) at 37 °C in 5% CO$_2$-in-air for 24 h as per previously described methods[63]. To prevent impaired T-cell activation, culture media was supplemented with 10% dialysed fetal bovine serum (FBS; Fisher Scientific) after 3 h. T-cells were treated with ML-226 (10 µM; Cambridge Bioscience), unless otherwise stated.

**Phenocopy assays.** For α-ketoglutarate dehydrogenase (α-KGDH) inhibition experiments, T-cells were activated in the presence and absence of CPI-613 (100 µM; Cambridge Bioscience). For ATP citrate lyase (ACLY) inhibition experiments, T-cells were activated in the presence and absence of BMS-303141 (1 µM) and sodium acetate (1 mM; both Merck). For lactate phenocopy experiments, T-cells were activated in the presence and absence of lactic acid (10 mM; Merck). For LXR activation experiments, T-cells were activated in the presence and absence of GW3965 (2 µM; Merck). For oxysterol treatment experiments, T-cells were activated in the presence and absence of 27-hydroxycholesterol or 24,25-epoxycholesterol (both 10 µM, Cambridge Bioscience). For LXR inhibition experiments, T-cells were activated in the presence and absence of GSK2033 (5 µM; Merck).

**Regulatory T-cell differentiation assay.** Human CD4+ naïve T-cells were cultured in AIM-V (ThermoFisher) medium supplemented with 50 IU/mL IL-2 (PeproTech), 1% penicillin/streptomycin and 10% Hyclone FBS at 37 °C in 5% CO$_2$-in-air for 6 d. T-cells were activated with plate bound anti-CD3 (1 µg/ml; OKT3; BioLegend) and soluble anti-CD28 (5 µg/ml; CD28.2; BioLegend). Regulatory T (Treg) cell polarisation conditions included TGFβ (5 ng/ml; Miltenyi), with Th0 cultured in the absence of TGFβ. T-cells were treated with ML-226 (10 µM) or DMSO vehicle control either at d0 of polarisation, or activated with plate bound anti-CD3 (2 µg/ml; OKT3; BioLegend) and soluble anti-CD28 (20 µg/ml; CD28.2; BioLegend) in the presence and absence of ML-226 following polarisation to Treg cells (Supplementary Fig. 8). Prior to intracellular FOXP3 staining, T-cells were activated with PMA (50 ng/ml) and ionomycin (500 ng/ml; both Merck) and protein transport blocked using Protein Transport Inhibitor Cocktail (eBioscience).

### Murine CD4+ T-cell isolation and culture
**Mice.** BDC2.5 TCR transgenic NOD mice (NOD.Cg-Tg(TcraBDC2.5,TcrbBDC2.5)1Doi/DoiJ; 004460) and Rag1-/-NOD mice (NOD.129S7(B6)-Rag1tm1Mom/J; 003729) were both purchased from Jackson Laboratory[64–66]. All mice received water and irradiated food (T.2919.CS; ENVIGO) ad libitum and were housed at Cardiff University in specific-pathogen-free Scantainers with 12 h light–dark cycles. All animal experiments were approved by the Cardiff University ethical review process and conducted under UK Home Office licence in accordance with the UK Animals (Scientific Procedures) Act 1986 and associated guidelines.

**In vitro T-cell differentiation assay.** Splenic CD4+ T-cells were isolated from C57BL/6 mice (aged 8–12 weeks) by MACS magnetic separation (Miltenyi) and cultured in RPMI-1640 (Gibco) media at 37 °C in 5% $CO_2$-in-air for 4 d. For Th17 polarising conditions, Iscove's Modified Dulbecco's Medium (IMDM; Gibco) was used for optimal Th17 differentiation. CD4+ T-cells were activated with plate-bound anti-CD3 (1 µg/ml; 145-2C11; R&D Systems) and soluble anti-CD28 (10 µg/ml; 37.51; eBioscience). Polarisation conditions were as follows: Th1 (IL-12 [20 ng/ml]), Th2 (IL-4 [40 ng/ml]), Th17 (TGFβ [1 ng/ml], IL-6 [20 ng/ml], IL-23 [20 ng/ml], anti-IL-2 [10 µg/ml, JES6–1A12]), Treg (TGFβ [5 µg/ml]). All cytokines were purchased from R&D Systems. T-cells were treated with WWL-222 (10 µM) or DMSO vehicle control.

**In vitro antigen presentation assay.** Splenic CD4+ T-cells were isolated from BDC2.5 NOD mice as per the manufacturer's instructions (MojoSort™ Mouse CD4 T-cell Isolation Kit; BioLegend). Splenic antigen-presenting cells (APCs) were depleted of T-cells through incubating cells with anti-Thy1 antibody (50 µg/ml, M5/49.4.1; BioX-cell) and 1:20 rabbit complement (Merck) over 1 h at 37 °C. $1 \times 10^5$ BDC2.5 CD4+ T-cells were co-cultured 1:1 with APCs in the presence of BDC2.5 hybrid insulin peptide (DLQTLALWSRMD)[34] for 48 h, in the presence or absence of WWL222 (10 µM), prior to harvest.

**In vivo diabetes adoptive transfer.** BDC2.5 CD4+ T-cells were activated and isolated as previously described. $5 \times 10^6$ BDC2.5 CD4+ T-cells were i.v. adoptively transferred into 3–4 week-old Rag1−/−NOD recipients. Recipient mice were injected with 2.5 mg/kg of WWL222 daily i.p. from the day of T-cell transfer until termination. Mice were screened for glycosuria daily (Diastix; Bayer) with diabetes confirmed by blood glucose measurement (>13.9 mmol/L).

$12 \times 10^6$ splenocytes from newly-diabetic mice were i.v. adoptively transferred into 3–4 week-old Rag1−/−NOD recipients. Recipient mice were injected with 2.5 mg/kg of WWL222 daily i.p. from the day of T-cell transfer until termination. Mice were screened for glycosuria daily (Diastix; Bayer) with diabetes confirmed by blood glucose measurement (>13.9 mmol/L).

## Cell lines
Jurkat E6.1 T-cells were obtained from Marc Mansour at University College London. Jurkat T-cells were maintained in RPMI-1640 supplemented with 10% fetal bovine serum, 2 mM glutamine and 100U penicillin per 100 µg/ml streptomycin (all Gibco) and routinely tested negative for mycoplasma infection. Cells were cultured at a density of $0.1 \times 10^6$ cells/mL and cell counts performed using a Countess 3 Automated Cell Counter (Invitrogen).

## CRISPR/Cas9
To generate ABHD11 knockdown clones, $1 \times 10^6$ Jurkat T-cells were nucleofected in SE buffer (Lonza) supplemented with 60 pM Cas9 (IDT) and 180 pM of two distinct ABHD11-specific single guide RNA (IDT; Supplementary Table 3) with 4D Nucleuofector X Unit (Lonza) set to CK116. Cells were left to recover overnight and single cell clones grown for two weeks. ABHD11 knockdown efficiency was tested by immunoblotting and different degrees of knockdown were noted for each of the clones analysed.

## Flow cytometry
### Gating strategy.
Flow cytometry was performed on T-cells following cell culture. Cell death was monitored using DRAQ7™ (1 µM, DR71000; Biostatus), unless otherwise stated, and dead cells were excluded from analysis. Cell doublets were also excluded from analysis. A representative gating strategy can be found as Supplementary Fig. 9.

### Surface staining.
For human T-cells, surface staining was performed at room temperature (RT) for 15 min in the dark. Antibodies were used as follows: anti-CD25 (Pacific Blue, mIgG1κ, BC96, 302627), anti-CD25 (PE-Vio 615, rhIgG1, REA570, 130-123-035; Miltenyi), anti-CD44 (FITC, rhIgG1, REA690, 130-113-341; Miltenyi), anti-CD44 (Pacific Blue, mIgG1κ, BJ18, 338823), anti-CD69 (APC, mIgG1κ, FN50, 310910), anti-CD69 (FITC, mIgG1κ, FN50, 310904). Antibodies were purchased from BioLegend, unless otherwise stated.

For murine splenic T-cells from C57BL/6 mice, cell death was monitored using the Zombie Aqua Fixable Viability Kit (423102; BioLegend).

For murine splenic T-cells from BDC2.5 TCR transgenic NOD mice for in vitro antigen presentation, cells were labelled with Vybrant CFDA-SE Cell Tracer Kit (0.5 µM; V12883; Invitrogen™) to assess proliferation. Single cell suspensions were incubated with TruStain FcX™ Fc Receptor Blocking Solution (clone 93; BioLegend) for 10 min at 4 °C prior to staining for surface markers for 30 min at 4 °C. Antibodies were used as follows: anti-CD4 (APC/Cy7, rIgG2bκ, GK1.5, 100414), anti-CD8a (PerCP/Cy5.5, rIgG2aκ, 53-6.7, 100734), anti-CD25 (APC, rIgG1λ, PC61, 102012), anti-CD26L (Pacific Blue™, rIgG2aκ, MEL-14, 104424) and anti-CD69 (PE/Cy7, ahIgG, H1.2F3, 104512). Antibodies were purchased from BioLegend. Cell death was monitored using the Zombie Aqua Fixable Viability Kit (423102; BioLegend). Proliferating BDC2.5 CD4 + T-cells (CFDA-SE^low) were analysed.

For murine splenic T-cells from BDC2.5 TCR transgenic NOD mice, single cell suspensions were incubated with TruStain FcX™ Fc Receptor Blocking Solution (clone 93; BioLegend) for 10 min at 4 °C prior to staining for surface markers for 30 min at 4 °C. Antibodies were used as follows: anti-CD4 (BUV496, LewIgG2b, GK1.5, 612952; BD), anti-CD8 (PerCP/Cy5.5, rIgG2aκ, 53-6.7, 100734), anti-CD11b (Brilliant Violet 510™, rIgG2bκ, M1/70, 101263), anti-CD11c (BUV661, ahIgG2, N418, 750449; BD), anti-CD19 (APC-Cy7, rIgG2aκ, 1D3, 152412), anti-CD25 (APC, rIgG1, PC61, 102012), anti-CD40 (BUV615, LouIgG2aκ, 3/23, 751646; BD), anti-CD44 (BUV805, rIgG2bκ, IM7, 741921; BD), anti-CD62L (BUV395, rIgG2aκ, MEL-14, 569400; BD), anti-CD69 (BUV563, ahIgG1λ3, H1.2F3, 612952; BD), anti-CD80 (AlexaFluor® 594, rrIgG, 2740B, FAB7401T-100ug; Bio-Techne), anti-CD86 (BUV737, rIgG2bκ, PO3, 741757; BD), anti-MHC-I-[H-2kd] (Brilliant Violet 421™, mIgG2aκ, SF1-1.1, 116623) and anti-MHC-II-[I-AK] (PE, mIgG2aκ, 10-3.6, 109908). Antibodies were purchased from BioLegend, unless otherwise stated. Cell death was monitored using either the Zombie Aqua Fixable Viability Kit (423102; BioLegend) or Fixable Viability Stain 575 V (565694; BD).

For murine splenocytes from NOD mice, ingle cell suspensions were incubated with TruStain FcX™ Fc Receptor Blocking Solution (clone 93; BioLegend) for 10 min at 4 °C prior to staining for surface markers for 30 min at 4 °C. Antibodies were used as follows: anti-CD4 (Brilliant Violet 650™, rIgG2bκ, GK1.5, 100469), anti-CD8 (PerCP/Cy5.5, rIgG2aκ, 53-6.7, 100734), anti-CD11b (APC, rIgG2bκ, M1/70, 101212), anti-CD11c (PE, ahIgG1λ2, HL3, 553802; BD), anti-CD19 (Alexa Fluor® 700, rIgG2aκ, 6D5, 115528), anti-CD25 (PE/Cy7, rIgG1λ, PC61, 102016), anti-CD44 (PerCP/Cy5.5, rIgG2bκ, IM7, 103032), anti-CD45 (APC/Cy7, rIgG2bκ, 30-F11, 103116), anti-CD62L (BV786, rIgG2aκ, MEL-14, 564109; BD). Antibodies were purchased from BioLegend, unless otherwise stated. Cell death was monitored using the Zombie Aqua Fixable Viability Kit (423102; BioLegend).

### Intracellular staining.
For human T-cells, intracellular markers were stained using the eBioscience™ Foxp3/Transcription Factor Staining Buffer Set (00-5523-00) and cell death monitored using DRAQ7™ (1 µM, DR71000; Biostatus) as per the manufacturer's instructions (both Invitrogen). Following surface staining, cells were fixed for 30 min at RT before staining in permeablisation buffer. Primary antibodies were incubated for 1 h at RT, followed by incubation with an anti-rabbit IgG secondary antibody (Brilliant Violet 421™, donkey pIg, Poly4064, 406410; BioLegend) for 30 min at RT. Primary antibodies used were purchased from Abcam, unless otherwise stated: anti-

Histone H3 (acetyl K9; H3K9ac, ab12178), anti-Histone H4 (acetyl K8; H4K8ac, ab15823) and anti-acetyl lysine (ab21623).

For human T-cell FOXP3 staining, intracellular markers were stained using the eBioscience™ Foxp3/Transcription Factor Staining Buffer Set (00-5523-00; Invitrogen) and cell death monitored using DRAQ7™ (1 µM, DR71000; Biostatus) as per the manufacturer's instructions (both Invitrogen). Following surface staining with anti-CD4 (Alexa Fluor® 647, mIgG2bκ, OKT4, 317422; BioLegend), cells were fixed for 30 min at RT before staining in permeablisation buffer. Cells were permeabilised overnight with anti-FOXP3 (PE, mIgG1κ, 206D, 320108; BioLegend).

For murine T-cells from C57BL/6 mice spleens, intracellular markers were stained using the Cytofix/Cytoperm Fixation/Permeabilisation kit (BD) as per the manufacturer's instructions. 4 h prior to intracellular staining, cells were activated with PMA (50 ng/ml) and ionomycin (500 ng/ml), and protein transport blocked using monensin (3 µM; all Merck). Following viability staining, cells were fixed and permeabilised and stained with either: anti-CD4 (Brilliant Violet 785™, rIgG2aκ, RM4-5, 100551; BioLegend), anti-FOXP3 (PE, rIgG2ακ, FJK-16S, 12-5773-82; eBioscience), anti-IFNγ (eFluor™ 450, rIgG1κ, XMG1.2, 48-7311-82; Invitrogen), anti-IL-13 (Alexa Fluor™ 488, rIgG1κ, eBio13A, 53-7133-82; Invitrogen) and anti-IL-17A (PE, rIgG1κ, TC11-18H10, 561020; BD Pharmigen); or anti-CD4 (PerCP/Cy5.5, rIgG2aκ, RM4-5, 45-0042-82; eBioscience), anti-FOXP3 (PE, rIgG2ακ, FJK-16S, 12-5773-82; eBioscience), anti-IFNγ (eFluor™ 450, rIgG1κ, XMG1.2, 48-7311-82; eBioscience), anti-IL13 (AlexaFluor™ 488, rIgG1κ, eBio13A, 53-7133-82; eBioscience) and anti-IL17A (Alexa Fluor® 647, rIgG1κ, TC11-18H10.1, 506912; BioLegend).

For murine T-cells from BDC2.5 TCR transgenic NOD mice spleens, intracellular markers were stained using the Cytofix/Cytoperm Fixation/Permeabilisation kit (BD) as per the manufacturer's instructions. 3 h prior to intracellular staining, cells were activated with PMA (50 ng/ml) and ionomycin (500 ng/ml), and protein transport blocked using GolgiPlug (BD). Following surface staining, cells were fixed for 20 min at RT before permeabilisation. Cells were incubated with TruStain FcX™ as previously described prior to staining for intracellular cytokines. Antibodies used were purchased from BioLegend, unless otherwise stated: anti-IL-2 (Spark Red™ 718, rIgG2bκ, JES6-5H4, 503852), anti-IL-6 (APC, rIgG1κ, MP5-20F3, 503852), anti-IL-10 (Brilliant Violet 605™, rIgG2bκ, JES5-16E3, 505031), anti-IL-12/23 (PE-Cy7, rIgG2κ, C15.6, 505210), anti-IL-17a (Brilliant Violet 786™, rIgG2κ, TC11-18H10.1, 506928), anti-IFNγ (Brilliant Violet 650™, rIgG1κ, XMG1.2, 505832) and anti-TNFα (Brilliant Violet 750™, rIgG1κ, MP6-XT22, 506358).

For murine splenocytes from NOD mice spleens, intracellular markers were stained using the Cytofix/Cytoperm Fixation/Permeabilisation kit (BD) as per the manufacturer's instructions. Three hours prior to intracellular staining, cells were activated with PMA (50 ng/ml) and ionomycin (500 ng/ml), and protein transport blocked using GolgiPlug (BD). Following surface staining, cells were fixed for 20 min at RT before permeabilisation. Cells were incubated with TruStain FcX™ as previously described prior to staining for intracellular cytokines. Antibodies used were purchased from BioLegend: anti-IFNγ (Brilliant Violet 605™, rIgG1κ, XMG1.2, 505840), anti-TNFα (Alexa Fluor® 488, rIgG1κ, MP6-XT22, 506313).

**Puromycin incorporation.** Protein translation was assessed using anti-puromycin (AlexaFluor® 488, 12D10, MABE343-AF488; Merck). Puromycin (10 µM; Merck) was added 15 min prior to the end of 4 h and 24 h T-cell activation. Cells were washed in ice-cold PBS before intracellular staining was performed using Inside Stain Kit (Miltenyi) as per the manufacturer's instructions. Cells were fixed for 20 min at RT, permeabilised for 15 min at RT, before staining for 1 h at 4 °C in permeabilisation buffer.

**Proliferation.** Proliferation was assessed using CellTrace™ CFSE Cell Proliferation kit (5 µM; C34570; ThermoFisher). Cells were stained for 20 min in the dark prior to 72 h activation. Proliferation-associated parameters were calculated using the FlowJo Proliferation Platform (TreeStar).

**Mitochondrial characteristics.** For mitochondria content and membrane potential, cells were incubated with MitoTracker™ Green FM (100 nM; M7514; ThermoFisher) or TMRE (50 nM; ab113852; Abcam) for 20 min at 37 °C, respectively.

**Reactive oxygen species.** For mitochondrial ROS staining, cells were incubated with MitoSOX™ Red (5 µM; M36008, ThermoFisher) for 20 min at 37 °C. For total ROS staining, cells were incubated with CellROX™ Green (5 µM; C10492; ThermoFisher).

**Purity.** Human CD4+ T-cell purity was monitored using anti-CD3 (Brilliant Violet 570™, mIgG1κ, UCHT1, 300436) and anti-CD4 (Alexa-Fluor® 647, mIgG2b, OKT4, 317422; BioLegend). CD4+ effector T-cell purity was monitored using anti-CD4 (AlexaFluor® 647, mIgG2b, OKT4, 317422; BioLegend), anti-CD45RA (Brilliant Violet 605™, mIgG2b, HI100, 304134; BioLegend), anti-CD45RO (FITC, mIgG2a, UCHL1, 304204; BioLegend) and anti-CD197 (Pacific Blue, mIgG2a, G043H7, 353210; BioLegend). Percentage purity was consistently >90%.

**Analysis.** Human T-cells were acquired on a Novocyte 3000 (Agilent). Murine T-cells were acquired on a Novocyte 3000 (Agilent), FACS Canto II or a Symphony A3 Cell Analyser (both BD). Analysis was performed using FlowJo version 10 (TreeStar).

### Enzyme linked immunosorbent assay
Cell-free supernatants were analysed for human IL-2 (DY202), IL-10 (DY217B), IL-17 (DY317), IFNγ (DY285B) and TNFα (DY210) by DuoSet ELISA as per the manufacturer's instructions (R&D Systems). 96-well half-area plates were coated with capture antibody overnight at 4 °C. Cell-free supernatants were diluted to an appropriate concentration and incubated for 2 h at RT with gentle agitation, followed by 2 h with the kit-specific detection antibody, and finally 20 min with streptavidin-horse radish peroxidase. The plate was then incubated with a 1:1 mixture of hydrogen peroxide and tetramethylbenzoic acid (555214; BD) at RT. Absorbance was measured at 450 nm following the addition of sulfuric acid (Merck) to each well. All values were corrected to the blank.

### Immunoblot
T-cells were lysed in PhosphoSafe Extraction Buffer (Merck). Cell lysate proteins were quantified, denatured and separated using SDS-PAGE. Polyvinylidene difluoride membranes were probed with antibodies targeting: ABHD11 (PA5-54962; Invitrogen), acetylated lysine (MA5-33031; Invitrogen), Bcl-2 (1507), Caspase-3 (9662), cleaved Caspase-3 (9661), phospo-LAT$^{Tyr220}$ (3584), LAT (45533), phospho-PLCγ$^{Tyr783}$ (14008), PLCγ (5690), phospho-ZAP70$^{Tyr493/Tyr526}$ (2704), ZAP70 (3165). All antibodies were purchased from Cell Signaling, unless otherwise stated, and used at a 1:1000 dilution. Protein loading was monitored using β-actin (ab8226; Abcam).

### Gene expression analysis
**RNA preparation.** RNA was extracted using RNeasy® Mini Kit columns (74014; Qiagen) as per the manufacturer's guidelines. RNA purity was assessed using a Nanodrop™ spectrophotometer and measured A260/280 and A260/230 ratios were typically between 1.8–2.2.

**qPCR.** cDNA was prepared from 500 ng of RNA using the High-Capacity cDNA Reverse Transcription Kit (4368813; ThermoFisher) as

per the manufacturer's instructions. qPCR reactions were prepared using Fast SYBR® Green Master Mix (4385612; ThermoFisher) as per the manufacturer's guidelines. All gene expression analyses were normalised to *RPL19*. Primers were purchased from Merck (VC0021N) and sequences used can be found in Supplementary Table 4.

## Metabolic analysis

Metabolic analysis was performed using the Seahorse Extracellular Flux Analyzer XFe96 (Agilent) following cell culture. T-cells were resuspended in RPMI phenol red free media supplemented with glucose (10 mM), glutamine (2 mM) and pyruvate (1 mM; all Agilent). T-cells were seeded onto a Cell-Tak (354240; Corning) coated microplate allowing the adhesion of T-cells. Mitochondrial and glycolytic respiratory parameters were measured using OCR (pmoles/min) and ECAR (mpH/min), respectively. Injections included: oligomycin (1 μM), FCCP (1 μM), rotenone (1 μM) and antimycin A (1 μM) and monensin (20 μM). All chemicals were purchased from Merck, unless otherwise stated.

## Liquid-chromatography mass-spectrometry (LC-MS)

T-cells were washed twice with ice-cold PBS and lysed in methanol, acetonitrile and water (v/v 5:3:2) following cell culture. Chromatographic separation of metabolite extracts was done using a ZIC-pHILIC column (SeQuant; 150 mm × 2.1 mm, 5 μm; Merck) and ZIC-pHILIC guard column (SeQuant; 20 mm × 2.1 mm; Merck) coupled to Vanquish HPLC system (ThermoFisher). A gradient programme was employed, using 20 mM ammonium carbonate (pH 9.2, 0.1% v/v ammonia, 5 μM InfinityLab deactivator (Agilent)) as mobile phase A and 100% acetonitrile as mobile phase B. Elution started at 20% A (2 min), followed by a linear increase to 80% A for 15 min and a final re-equilibration step to 20% A. Column oven was set to 45 °C and flow rate to 200 μl min$^{-1}$.

Metabolite profiling and identification was achieved using a Q Exactive Plus Orbitrap mass spectrometer (ThermoFisher) equipped with electrospray ionization as per previously described methods[67]. Polarity switching mode was used with a resolution (RES) of 70,000 at 200 m/z to enable both positive and negative ions to be detected across a mass range of 75 to 1000 m/z (automatic gain control (AGC) target of $1 \times 10^6$ and maximal injection time (IT) of 250 ms).

Data analysis was undertaken in Skyline (version 23.1.0.455)[68]. Identification was accomplished by matching accurate mass and retention time of observed peaks to an in-house library generated using metabolite standards (mass tolerance of 5 ppm and retention time tolerance of 0.5 min).

## Stable isotope tracer analysis (SITA) by LC-MS

Isolated human CD4+ T-cells were incubated with universally heavy-labelled $^{13}C_6$-glucose (11.1 mM; Cambridge Isotopes) in glucose-free RPMI (Gibco), or $^{13}C_5$-glutamine (2 mM; Cambridge Isotopes) in glutamine-free RPMI (Gibco), and activated in the presence and absence of ML-226, as previously described. T-cells were then washed twice with ice-cold PBS and lysed in methanol, acetonitrile and water (v/v 5:3:2). Metabolite extraction and subsequent LC-MS analysis were performed as previously described. For tracing analysis, integration of each isotopologue was manually verified.

## Lipid analysis by LC-MS/MS

Aliquots corresponding to $2.5 \times 10^5$ cells were transferred to fresh glass tubes for liquid-liquid extraction (LLE). Samples were extracted using a modified methyl tert-butyl ether (mTBE) method[69]. Briefly, 1 ml water, 1 ml methanol and 2 ml mTBE were added to the glass tubes containing the sample. The mixture was vortexed and centrifuged at 2671×*g* for 5 min. The organic phase was collected to fresh glass tubes and spiked with 20 μL of a 1:5 diluted Splash Lipidomix standards mixture (Avanti Polar Lipids, Alabaster, AL). Samples were dried under $N_2$ and resuspended in 400 μl of hexane.

Lipids were analysed by LC-MS/MS using a SCIEX QTRAP 6500+ (SCIEX, Framingham, MA) equipped with a Shimadzu LC-30AD (Shimadzu, Columbia, MD) HPLC system and a 150 × 2.1 mm, 5 μm Supelco Ascentis silica column (Supelco). Samples were injected at a flow rate of 0.3 ml min$^{-1}$ at 97.5% Solvent A (hexane) and 2.5% solvent B (mTBE). Solvent B was increased to 5% over 3 min and then to 60% over 6 min. Solvent B was decreased to 0% during 30 s while Solvent C (9:1 [v/v] isopropanol-water) was set at 20% and increased to 40% during the following 11 min. Solvent C is increased to 44% over 6 min and then to 60% over 50 s. The system was held at 60% solvent C for 1 min prior to re-equilibration at 2.5% of solvent B for 5 min at a 1.2 ml min$^{-1}$ flow rate. Solvent D (95:5 [v/v] acetonitrile-water with 10 mM ammonium acetate) was infused post-column at 0.03 ml min$^{-1}$. Column oven temperature was 25 °C. Data were acquired in positive and negative ionization mode using multiple reaction monitoring (MRM). The LC-MS/MS data was analyzed using MultiQuant software (SCIEX). The identified lipid species were normalized to its corresponding internal standard.

## Sterol analysis

T-cell pellets (>4 × 10$^6$ cells) were snap-frozen in liquid nitrogen. Sterol extraction, hydrolysis, isolation and analysis was performed according to previously described methods[70].

## RNA-Seq

**Sample preparation.** For human T-cells, pellets were snap-frozen in liquid nitrogen. RNA was extracted using RNeasy® Mini Kit columns (74014; Qiagen) as per the manufacturer's guidelines. For each sample, 2 μg of total RNA was then used in Illumina's TruSeq Stranded mRNA Library kit (#20020594). Libraries were sequenced on Illumina NovaSeq 6000 as paired-end 150-nt reads.

For Jurkat T-cells, a total of 200 ng of total RNA was used as input for the Oxford Nanopore Technologies (ONT) PCR cDNA barcoding library preparation protocol (SQK-PCB114.24). Briefly, RNA was reverse-transcribed using a strand-switching method, and cDNA was amplified with barcoded primers, employing a 4 min extension time and 14 PCR cycles. The barcoded cDNA was quantified and pooled in equimolar amounts before the addition of rapid sequencing adapters (RA). Libraries (25 fmol) were loaded onto R10.4.1 flow cells and sequenced on a PromethION P2 device. Data acquisition was conducted using MinKNOW, and basecalling was performed with Dorado (v0.8.0).

**Analysis.** For human RNA-Seq analysis, Fastq files were quality assessed and trimmed using FastP(v0.23.1)[71], before reads were mapped to the genome GRCh38 using STAR (Spliced Transcripts Alignment to a Reference; v2.7.9a)[72] with 2-pass method and multimapping set to 1. Featurecounts(v2.0.3)[73] was used to generate count files for each sample, with counting performed at the gene level. Differential gene expression analysis was calculated via eBayesian fit of TMM (Trimmed Mean of *M*-values) using an EdgeR workflow[74] of Limma-Voom(v3.58.1)[75]. Genes were filtered for an adjusted *p*-value < 0.05 and over representation analysis was performed using gProfiler (v.e111.eg58.p18.f463989d)[76] for Gene Ontology terms for Biological Processes (GO:BP). ReViGo(v1.8.1)[77] was used to reduce the repetition of canonical pathway terms. GeneSet enrichment analysis was performed using genekitR (v1.2.5)[78]. Protein-protein interaction was assessed using StringDB(v12.0)[79], whereby differentially expressed genes were treated as if they were fully transcribed into proteins. Transcription factor enrichment analysis was performed using X2Kweb (v14/22 Appyter)[80], which infers upstream regulatory networks from the differentially-expressed gene signature. For all enrichment analyses, differentially-expressed genes were separated into up and downregulated sets.

For RNA-Seq analysis on ABHD11 knockdown clones, sequencing data were processed using the nf-core/nanoseq pipeline (v3.1.0)[81] with

the cDNA protocol. Demultiplexing was skipped, and reads were aligned to the human reference genome (Gencode Release 45, GRCh38.p14) using minimap2. Transcript quantification was performed with bambu. Differential expression analysis was conducted in R (v4.3.2) using DESeq2 (v1.42.1). Data visualisations and exploratory analyses were performed with tidyverse (v2.0.0) and clusterProfiler (v4.10.1).

## α-ketoglutarate dehydrogenase activity assay

T-cell α-ketoglutarate dehydrogenase activity was measured following culture using the α-Ketoglutarate Dehydrogenase Activity Colorimetric Assay Kit (MAK189; Merck) as per the manufacturer's instructions. T-cells were lysed with the α-KGDH Assay Buffer provided, before commencing the reaction with α-KGDH developer and α-KGDH substrate. Absorbance was measured at 450 nm until the absorbance value for the most active sample exceeded that of the highest NADH standard concentration. All values were corrected to the blank.

## Lactate assay

Extracellular L-lactate concentrations were measured using L-Lactate Assay Kit I (Eton Bioscience, USA) as per the manufacturer's instructions. Cell-free supernatants were diluted to an appropriate concentration prior to addition of L-Lactate Assay Solution and incubation at 37 °C for 30 min. Absorbance was measured at 490 nm after the addition of acetic acid (0.5 M; Merck) to each well. All values were corrected to the blank.

## Statistical analysis

Statistical analysis was performed using GraphPad Prism version 10 (USA). Data are represented as the mean ± or + standard error of the mean (SEM). The one-sample Kolmogorov-Smirnoff test was used to test normality. Where no substantial deviations from normality were observed, it was considered appropriate to use parametric statistics. All experiments have biological replicate sample sizes of $\geq n = 3$ (exc. human T-cell proliferation experiment) and significant values were taken as $p \leq 0.05$.

## Ethics statement

This research complies with all relevant ethics regulations at Swansea University. For healthy controls, informed written consent was obtained from all study participants and ethical approval was obtained from Swansea University Medical School Research Ethics Committee (SUMSRESC; 2022-0029). For rheumatoid arthritis (RA) patient samples, informed written consent was obtained from all study participants and ethical approval was obtained from St. Vincent's University Hospital Ethics Committee (RS18-055). For type 1 diabetes (T1D) patient samples, informed written consent was obtained from all study participants and ethical approval was obtained from the Wales Research Ethics Committee (12/WA/0033).

## Reporting summary

Further information on research design is available in the Nature Portfolio Reporting Summary linked to this article.

## Data availability

All data are available upon request and can be found within the paper and supplementary information. The raw RNA-Seq data generated in this study have been deposited in the European Genome-phenome Archive (EGA) under the accession code EGAS50000001297. Mass spectrometry data are available through https://massive.ucsd.edu/ (dataset: MSV000099454). Source data are provided with this paper.

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

## Acknowledgements

We thank W. J. Griffiths for useful discussion, and all blood donors for their contribution to this work. We would also like to acknowledge Active Motif for their support. Images provided by and adapted from Servier Medical Art (https://smart.servier.com/) are licensed under Creative Commons Attribution 4.0. Y.R.J. is funded by a Swansea University Research Excellence Scholarship. A.H.U., F.B. and D.S. are funded by Cancer Research UK core funding to the CRUK Scotland Institute. L.C.D. is funded by an MRC New Investigator Research Grant (MR/Y013816/1). J.A.N is supported by a Wellcome Senior Clinical Research Fellowship (215477/Z/19/Z). J.V.V. is supported by a Cancer Research UK—Career Development Fellowship (RCCCDF-Nov23/100001) and by a Lord Kelvin/Adam Smith (LKAS) Leadership Fellowship from the University of Glasgow. J.G.M. is funded by National Institutes for Health (NIH) research grants (UL1TR003163, 1P30DK127984 and P01HL160487). G.W.J. is funded by a Versus Arthritis Career Development Fellowship (20305). J.A.P is supported by an MRC Career Development Award (MR/T010525/1). E.E.V is supported by a Diabetes UK RD Lawrence Fellowship (17/0005587), an MRC Research Grant (MR/Z505651/1) and by Cancer Research UK (C18281/A29019). N.J. is supported by an MRC New Investigator Research Grant (MR/X000095/1).

## Author contributions

B.J.J., Y.R.J., F.M.P-G., C.M., M.D.H., A.H.U., F.B., S.E., A.B., J.D., J.B., M.W., M.M., G.D.V., J.A.P., and N.J. performed the experiments. B.J.J, Y.R.J, D.K.F., L.V.S., A.E.H., I.R.H., D.S., J.V.V., G.W.J., J.A.P., E.E.V., and N.J. designed the experiments. B.J.J., Y.R.J., I.A.P., C.M., M.D.H., A.H.U., S.E., D.S., J.V.V., G.D.V., J.A.P., and N.J. analysed the data. B.J.J, Y.R.J., I.A.P. and N.J. visualised the data. A.H., D.J.V., U.F., and J.A.P. provided access to clinical samples. B.J.J., Y.R.J., J.G.C., J.B., L.C.D., M.N., D.K.F., L.V.S., B.F.C., A.E.H., J.A.N., U.F., D.S., J.V.V., J.G.M., G.W.J., J.A.P., E.E.V. and N.J. provided intellectual discussion. B.J.J., Y.R.J, J.A.P, E.E.V. and N.J. wrote the paper. All authors critically revised and approved the paper.

## Competing interests

M.N. is an employee of Lundbeck. The authors declare no competing interests.

## Additional information

¹Institute of Life Science, Swansea University Medical School, Swansea University, Swansea, UK. ²Cellular and Molecular Medicine, University of Bristol, Biomedical Sciences Building, Bristol, UK. ³Cancer Research UK Scotland Institute, Garscube Estate, Switchback Road, Glasgow, UK. ⁴Kathleen Lonsdale Institute for Human Health Research, Maynooth University, Maynooth, Co, Kildare, Ireland. ⁵Diabetes Research Group, Division of Infection and Immunity,

School of Medicine, Cardiff University, Cardiff, UK. [6]The Francis Crick Institute, 1 Midland Road, London, UK. [7]University of Montreal, Maisonneuve-Rosemont Hospital Research Centre, Montreal, Canada. [8]Division of Infection and Immunity/Systems Immunity University Research Institute, School of Medicine, Cardiff University, Cardiff, UK. [9]EULAR Centre of Excellence, Centre for Arthritis and Rheumatic Diseases, St Vincent's University Hospital, Dublin, Ireland. [10]Department of Chemistry, Scripps Research, La Jolla, CA, USA. [11]School of Biochemistry and Immunology, Trinity Biomedical Sciences Institute, Trinity College Dublin, Dublin, Ireland. [12]Division of Cell Signalling and Immunology, School of Life Sciences, University of Dundee, Dundee, UK. [13]Cambridge Institute of Therapeutic Immunology & Infectious Disease (CITIID), Jeffrey Cheah Biomedical Centre, Department of Medicine, University of Cambridge, Cambridge, UK. [14]Molecular Rheumatology, School of Medicine, Trinity Biomedical Sciences Institute, Trinity College Dublin, Dublin, Ireland. [15]School of Cancer Sciences, Wolfson Wohl Cancer Research Centre, University of Glasgow, Glasgow, UK. [16]Center for Human Nutrition, Department of Molecular Genetics, University of Texas Southwestern Medical Center, Dallas, TX, USA. [17]School of Translational Health Sciences, Dorothy Hodgkin Building, University of Bristol, Bristol, UK. [18]Integrative Epidemiology Unit, School of Population Health Science, University of Bristol, Bristol, UK. [19]These authors contributed equally: Benjamin J. Jenkins, Yasmin R. Jenkins. [20]These authors jointly supervised this work: James A. Pearson, Emma E. Vincent, Nicholas Jones. ✉e-mail: n.jones@swansea.ac.uk

