## [Transparent Peer Review file · Nature Communications]

Mitochondrial ABHD11 inhibition drives sterol metabolism to modulate T-cell effector function

Corresponding Author: Dr Nicholas Jones

Version 0:

Reviewer comments:

Reviewer #1

(Remarks to the Author)

Despite the authors agreeing that “the ABHD11 inhibition does not selectively target autoreactive IL-17, IFN γ , or TNF α -producing T cells, but rather impacts T-cells activated via TCR engagement regardless of subset” despite the fact that they recognize that “in the adoptive transfer model of T1D the observed delay in disease onset is modest” despite their acknowledgment that “Treg are affected by ABHD11 inhibition” they keep insisting that ABHD11 targeting is therapeutically important for autoimmune diseases.

The presented data do not support this conclusion. What the data support is the relevance of ABHD11 in TCR activation.

Reviewer #2

(Remarks to the Author)

REVIEWERS' COMMENTS

Reviewer #1 (Remarks to the Author):

Despite the authors agreeing that “the ABHD11 inhibition does not selectively target autoreactive IL-17, IFN γ , or TNF α -producing T cells, but rather impacts T-cells activated via TCR engagement regardless of subset” despite the fact that they recognize that “in the adoptive transfer model of T1D the observed delay in disease onset is modest” despite their acknowledgment that “Treg are affected by ABHD11 inhibition” they keep insisting that ABHD11 targeting is therapeutically important for autoimmune diseases. The presented data do not support this conclusion. What the data support is the relevance of ABHD11 in TCR activation.

We agree that our data robustly demonstrate a central role for ABHD11 in TCR signalling and general T-cell activation, and we do not dispute that ABHD11 inhibition affects multiple T-cell subsets, including Tregs and effector T-cells. As the Reviewer notes, ABHD11 targeting does not selectively inhibit only autoreactive or pro-inflammatory T-cells such as those producing IL-17, IFN γ , or TNF α . However, we respectfully maintain that these findings still support the therapeutic potential of ABHD11 inhibition in autoimmune disease contexts, and we would like to clarify the rationale for this.

First, dampening overall TCR-mediated activation (as ABHD11 inhibition achieves) can be clinically beneficial in autoimmune diseases, where chronic or inappropriate T-cell activation drives pathology. Indeed, many currently approved immunosuppressive therapies (e.g., calcineurin inhibitors) exert broad effects on T-cell activation without selectively sparing Tregs or targeting only pathogenic clones. Our data show that ABHD11 inhibition suppresses T-cell activation across subsets, and in doing so, attenuates disease onset in the adoptive transfer model of T1D, albeit modestly. We believe this supports proof-of-concept, especially given the aggressive nature of this model and the fact that many early-stage therapeutic candidates show partial efficacy at this stage of development.

Second, while Treg function is impacted, our results do not suggest complete loss of regulation. In fact, we observe a net immunosuppressive outcome, which leads to delayed autoimmune pathology. We acknowledge that future studies will be necessary to more precisely dissect how ABHD11 targeting influences the balance between effector and regulatory responses *in vivo*, and we have tempered the language in the revised manuscript accordingly:

‘In particular, we recognise that assessing Treg cell function in vivo is important in further understanding the therapeutic potential of ABHD11 inhibition, and is therefore a limitation of this current study.’

To address the Reviewer’s concern, we have modified the manuscript title, toned down certain statements throughout the manuscript and have revised the conclusion as follows:

‘Collectively, our study demonstrates that ABHD11 is a critical regulator of TCR signalling across T-cell subsets. Whilst ABHD11 inhibition does not selectively target autoreactive T-cells – similarly to other clinically available agents⁶² – the broad suppression of T-cell activation is sufficient to delay diabetes development in a preclinical model of autoimmunity.’

These findings support the immunomodulatory potential of targeting ABHD11 activity in T-cell biology.'

We hope this clarification addresses the Reviewer's concern and demonstrates that, while we do not claim selective targeting of pathogenic T-cells, our data support the therapeutic potential of ABHD11 inhibition as a general T-cell–modulating strategy.